# Satellite-based regional Sea Surface Salinity maps for enhanced understanding of freshwater fluxes in the Southern Ocean

Verónica González-Gambau[1], Estrella Olmedo[1], Aina García-Espriu[1], Cristina González-Haro[1], Antonio Turiel[1], Carolina Gabarró[1], Alessandro Silvano[2], Aditya Narayanan[2], Alberto Naveira-Garabato[2], Rafael Catany[3], Nina Hoareau[1], Marta Umbert[1], Giuseppe Aulicino[4], Yuri Cotroneo[4], Roberto Sabia[5], and Diego Fernández-Prieto[5]

[1]Barcelona Expert Center (BEC) and Institute of Marine Sciences (ICM), CSIC, P. Marítim de la Barceloneta, 37-49, 08003 Barcelona, Spain
[2]Ocean and Earth Science, University of Southampton, National Oceanography Centre, Southampton, United Kingdom
[3]Albavalor, SL, Calle Catedrático Agustín Escardino, 9, 46980 Paterna, Valencia, Spain
[4]Department of Science and Technologies, Università degli Studi di Napoli Parthenope, Naples, 80143, Italy
[5]European Space Agency, ESA-ESRIN. Largo Galileo Galilei 1 Casella Postale 64 00044, Frascati, Italy

**Correspondence:** Verónica González-Gambau (vgonzalez@icm.csic.es)

**Abstract.** This paper presents a newly developed Sea Surface Salinity (SSS) product for the Southern Ocean (SO), derived from SMOS (Soil Moisture and Ocean Salinity) measurements by the Barcelona Expert Center (BEC). The primary challenges in retrieving SSS from L-band brightness temperature (TB) measurements in the Southern Ocean include degraded sensitivity in cold waters, radiometric signal contamination near sea ice edges and low variability in SSS across the region. To address these challenges, significant improvements were made to the retrieval algorithms. The BEC SO SSS product v1.0 delivers 9-day SSS maps on a 25 km EASE-SL grid, generated daily. The time series spans from February 1, 2011, to March 31, 2023, with spatial coverage below 30ºS (https://doi.org/10.20350/digitalCSIC/15493).

The product shows high accuracy farther than 150 km from sea ice edges, with nearly zero bias and a standard deviation of 0.22 (compared to marine mammal data) and 0.25 (compared to TSG data from research vessels). Larger errors are observed within 150 km from the ice edges, due to residual sea-ice contamination and sampling-related errors in these dynamic areas. The product effectively captures seasonal and interannual variability, in line with the SOSE regional model. Although differences between satellite-derived and in situ salinity are more pronounced in these regions, the satellite product successfully reproduces the dynamics near ice edges.

This product will significantly contribute to the understanding of processes influenced by upper-ocean salinity, including sea ice dynamics, particularly, the reduction of Antarctic sea ice extent and the opening of offshore polynyas.

## 1  Introduction

Although the Southern Ocean (SO) represents less than one-third of the global ocean, it is responsible for absorbing 43% of the total oceanic anthropogenic $CO_2$ and 75% of the ocean's heat (Frölicher et al., 2015). Additionally, the heat stored in the SO is the primary direct and indirect (by its influence on winds and air temperature) driver of the Antarctic ice sheet melting (Holland

et al., 2020). Therefore, changes in the SO can have global consequences, both in terms of global warming and atmospheric carbon storage and sea level rise.

Observations in the SO have revealed rapid changes in recent years, including ocean warming and freshening (Swart et al., 2018), a reduction in sea ice extent (Purich and Doddridge, 2023; Eayrs et al., 2021), the reappearance of the Weddell Polynya (an ice-free area within the sea ice zone in the Weddell Sea) (Campbell et al., 2019) and increased melting of the Antarctic ice sheet (The IMBIE team, 2018). These changes have significant and profound impacts on the Earth's climate. However, the driving factors behind these changes remain unclear, as collecting measurements from this remote and harsh region is challenging, and modeling the complex interactions between the ocean, ice, and atmosphere is equally difficult.

Salinity and freshwater fluxes play a crucial role in these processes. Surface freshwater input from sea ice melting controls ocean circulation near the sea ice edge (Abernathey et al., 2016). In this region, atmospheric heat and carbon are absorbed by the ocean and sequestered at mid depths (500 to 1000 m) for decades. Near the Antarctic coast, the formation of sea ice during winter releases salt into the ocean. This process increases the density of seawater, causing it to sink into the abyss, to depths exceeding 3000 meters. This mechanism enables the storage of heat and carbon in the deep ocean for centuries. Finally, salinity influences the circulation at high latitudes due to its greater impact (at these latitudes) on ocean density compared to temperature (Roquet et al., 2022). As a result, variations in salinity near Antarctica can have a significant impact on ocean circulation and stratification (Klocker et al., 2023), and on heat transport toward the Antarctic ice sheet, influencing sea level rise (Silvano et al., 2018).

Satellite-observed Sea Surface Salinity (SSS) is a useful tool for understanding the drivers of these changes. In Garcia-Eidell et al. (2019), four global satellite SSS products were compared to assess the consistency of SSS distributions in the SO. The study revealed discrepancies between the products available at that time, particularly in terms of interannual and seasonal SSS variations. These discrepancies arose from the significant challenges involved in retrieving satellite SSS in polar regions. These include the contamination of the radiometric signal near sea ice transitions and the low brightness temperature (TB) sensitivity to SSS changes in cold waters (decreasing from 0.5 to 0.3 as sea surface temperature declines from 15°C to 5°C (Yueh et al., 2001)). The BEC team developed a specific product for the Arctic (Martínez et al., 2022). However, there is a significant difference between the Arctic and Southern Oceans. In the Southern Ocean, the SSS variability is notably lower than in the Arctic. This highlights the need to maximize the signal-to-noise ratio, particularly through enhanced Level 1 algorithms to improve TB quality. Additionally, signal contamination near sea ice is a key aspect to be specifically addressed in the SO by correcting SSS biases based on the distance from the sea ice edges.

Given these challenges, there is a clear need for the development of regional satellite SSS products specifically designed for the SO.

In this context, BEC L3 SO SSS product v1.0 has recently been developed for the SO (González-Gambau et al., 2023) and is presented in this paper. The key modifications to the conventional salinity retrieval algorithms used in generating this tailored SSS product focus on two main aspects: (i) reducing the contamination of the SSS signal near sea ice transitions, and (ii) minimizing radiometric errors to improve SSS accuracy, given the region's low SSS variability. This newly developed SSS product is a crucial tool for advancing our understanding of the Southern Ocean climate system and its ongoing changes.

Specifically, it will provide a solid foundation for addressing critical questions, such as the role of freshwater fluxes in shaping SO circulation and provide additional information on the various drivers behind the recent decline in Antarctic sea ice.

The article is structured as follows: Section 2 describes the data and algorithms used for the development of the regional SSS product. Section 3 presents the datasets and the metrics that have been used in the quality assessment, and discusses the quality of the satellite SSS product. Section 4 summarizes the main conclusions of this study. Section 5 contains the instructions to access the data.

## 2 BEC L3 SO SSS product v1.0 development

This section describes the datasets and the key algorithms used in the development of the BEC SO SSS product from observations of the SMOS (Soil Moisture and Ocean Salinity) mission.

### 2.1 Datasets

**SMOS Level 0 data**

The TBs are derived from the ESA SMOS Level 0 (L0) data. L0 contains the raw satellite data, both telemetry and observation data. This data has been downloaded from the SMOS mission Data Processing Ground Segment (DPGS) at BEC.

**Auxiliary data for SSS retrieval**

- *Data for geophysical corrections:* This data includes the geophysical parameters required to compute radiative and roughness corrections (Zine et al., 2008) and, in some cases, to filter out invalid data points. The data is provided by ECMWF (European Centre for Medium-Range Weather Forecasts) for each SMOS overpass (Sabater and De Rosnay, 2010) (https://smos-diss.eo.esa.int/oads/access/collection/AUX_Dynamic_Open). The dataset includes sea ice cover, rain rate, 10-meter wind speed, 10-meter neutral equivalent wind (both zonal and meridional components), significant wave height of wind waves, 2-meter air temperature, surface pressure, and vertically integrated total water vapor. Although the dataset also contains sea surface temperature (SST), a more detailed and specific analysis is performed for this variable (see below).

- *Sea Surface Temperature (SST).* We evaluated the quality of several SST products to find the one with the best performance in the SO region: AMSR2 (Wentz et al., 2021), CMC (Brasnett, 2008), OSTIA (Good et al., 2020), CCI (Merchant et al., 2019) and MUR (Chin et al., 2017). The assessment was based on three key aspects: (i) comparison with ARGO floats, (ii) spectral analysis to assess the effective spatial resolution (details provided in Section 3.2 of (Olmedo et al., 2021a)), and (iii) singularity spectrum analysis to assess the structural coherence and dynamical quality of the products (González-Haro et al., 2024). Among the analyzed products, the GHRSST Level 4 MUR Global Foundation SST Analysis (v4.1) (NASA/JPL, 2015) demonstrates the best performance in the SO (according to the chosen criteria). This product is derived from nighttime observations collected by multiple sensors (microwave and infrared radiometers) and

iQuam in situ observations. This SST product has been re-gridded from its native resolution ($0.01°$) to its effective spatial resolution, determined by spectral analysis ($0.1°$ grid).

- *Annual salinity climatology*. We use the value provided in the average decadal product by the World Ocean Atlas 2013 (WOA2013) at $0.25° \times 0.25°$ (Levitus et al., 2014) as the multiyear salinity reference (see section 2.2.3). A nearest-neighbor interpolation is applied to compute the climatology at the product grid (EASE-SL 25 km).

**Data for SSS filtering and correction**

- *Sea Ice Concentration (SIC)*. We use the EUMETSAT OSI SAF (Ocean and Sea Ice Satellite Application Facility) global sea ice concentration interim climate data record (v3.0, 2022), OSI-430-a, doi:10.15770/EUM_SAF_OSI_0014, available at https://osi-saf.eumetsat.int/products/osi-430-a). This dataset is employed for three purposes: (i) enhancing the quality of brightness temperatures near ice edges (as discussed in Section 2.2.2), (ii) excluding SSS retrievals in ice-covered regions by filtering out retrievals where sea ice is present (further details can be found in Section 2.2.3), and (iii) characterizing and correcting systematic biases on SSS depending on the distance from sea ice.

- *Argo floats*. We use in situ salinity data from Argo floats (Argo, 2025) for correcting temporal biases in SSS maps (see Section 2.2.4. These measurements can be downloaded from ftp://ftp.ifremer.fr/ifremer/argo. In computing this correction, only the uppermost Argo salinities at a depth between 5 and 10 m are considered. More details about the filtering criteria can be found in (Olmedo et al., 2021a).

## 2.2 Algorithm developments

### 2.2.1 BEC SMOS data processing chain

This section outlines the main steps of BEC SMOS data processing chain and highlights the key differences in the algorithms used to generate earlier BEC SSS products, including the Arctic SSS v3.1 (Martínez et al., 2022), the Global SSS v2.0 (Olmedo et al., 2021a) and the Baltic SSS v1.0 (González-Gambau et al., 2022).

The BEC SMOS data processing chain is able to ingest both ESA Level 0 (L0, raw data) and ESA Level 1B data (L1B, Fourier coefficients of the brightness temperatures) to derive the brightness temperatures at the antenna reference frame for each snapshot. In the case of SO, we generate the SMOS TB from the L0 data because some critical algorithms steps and corrections, not considered in the ESA operational TBs (v724) (Oliva et al., 2020), are needed to enhance TB accuracy in the SO region. The data processing chain for the generation of the BEC SO SSS product is shown in Fig. 1. For the generation of TB we use the MIRAS Testing Software (MTS) (Corbella et al., 2008). We apply the so-called ALL-LICEF calibration approach (Corbella et al., 2016), since it improves the consistency between the zero-baseline measurements and the remaining visibilities. The combined use of the ALL-LICEF calibration and the $G_{kj}$ correction for residual calibration errors (Corbella et al., 2015) has been demonstrated to significantly reduce contamination, especially in areas near land-sea and ice-sea boundaries (González-Gambau et al., 2017; González-Gambau et al., 2022). The brightness temperatures are reconstructed from the normalized visibilities (Corbella et al., 2019). To minimize radiometric errors, we apply the nodal sampling technique to the brightness

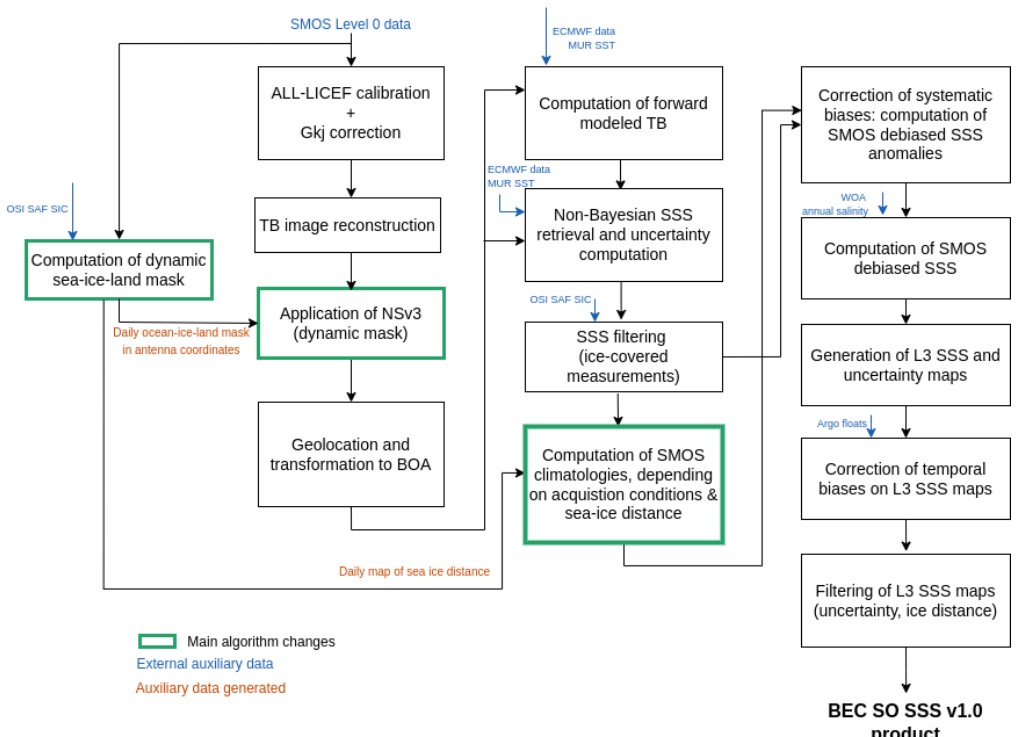

**Figure 1.** Block diagram of the BEC SO SSS processor.

temperatures (González-Gambau et al., 2016). A key modification has been incorporated into this algorithm to further enhance the quality of the SMOS observations in the SO, as detailed in Section 2.2.2. This is one of the key differences compared to Arctic SSS v3.1 and Global SSS v2.0, which use ESA L1B files (v620, from the previous reprocessing). Hence, neither the
120 ALL-LICEF and $G_{kj}$ correction nor the nodal sampling were applied. In the case of Baltic SSS v1.0, processing started from L0 and only the ALL-LICEF and $G_kj$ correction were applied.

These TBs are then geolocalized and transformed from TOA (Top of the Atmosphere) to BOA (Bottom of the Atmosphere), as detailed in (González-Gambau et al., 2017).

The difference between the SMOS-measured and the modeled First Stokes parameter is minimized to retrieve SSS. The
125 modeled TBs are obtained by using the geophysical model function presented in (Zine et al., 2008). The geophysical model that relates the modeled TB to SSS relies, unlike the official processor, on the Meissner and Wentz dielectric constant model (Meissner and Wentz, 2012), by accounting for the radiometric errors as developed for the Baltic Sea (González-Gambau et al., 2022). This is another important difference compared to Arctic SSS v3.1, which uses directly the Meissner and Wentz dielectric constant model (Meissner and Wentz, 2012) and compared to the Global SSS v2.0 product, which uses the Klein and Swift
dielectric constant model (Klein and Swift, 1977).

To model the roughness component, the semiempirical roughness model derived by (Guimbard et al., 2012) has been used. The contributions of other main sources, such as the reflected emission of the atmosphere, the reflection on the sea surface of the galactic emission, and the Sun glint, have also been corrected for. More details about these corrections can be found in (Olmedo et al., 2021a). A SSS value is retrieved for each TB measurement, unlike the conventional Bayesian approach, which retrieves a single SSS by considering all multi-angular TBs along the same dwell line. These raw SSS data are then filtered and combined to generate SSS maps. The Debiased non-Bayesian (DNB) methodology is used to reduce systematic SSS biases (Olmedo et al., 2017). The underlying hypothesis is that these systematic biases are the same for all the raw SSS values acquired under fixed acquisition conditions. Thus, measurements collected under the same conditions can be aggregated to compute the typical SSS value (hereafter referred to as SMOS-based climatologies) that can be used to correct systematic SSS biases , providing the debiased non-Bayesian SMOS anomalies. The way the raw SSS are classified to compute the SMOS-based climatologies for fixed acquisition conditions has been specifically tailored for each BEC SSS product. In the Global SSS v2.0, acquisition conditions were defined by longitude-latitude coordinates, orbit direction, and antenna position. For the Arctic SSS v3.1, systematic bias correction was applied to TB instead of SSS. However, we observed that residual errors were lower when the debiasing was applied to SSS. In the Baltic SSS v1.0, the debiasing was also applied to SSS, but with an additional variable in the classification of SSS: the SST. For the Southern Ocean, the modification is driven by the need to reduce systematic biases close to ice margins. The changes introduced at the SSS level for the SO are detailed in section 2.2.3.

### 2.2.2 Reduction of TB radiometric errors: Application of Nodal Sampling with a dynamic sea-ice mask

To capture the low SSS variability in the SO, it is essential to minimize TB radiometric errors as much as possible. In this regard, we have modified the Nodal Sampling (NS) methodology to enhance its performance near sea ice edges. Initially developed to mitigate contamination from Radio-Frequency Interferences (RFI) and sharp TB changes (González-Gambau et al., 2015), NS not only reduces RFI tails and ripples, but also effectively decreases general radiometric noise by approximately 50% (González-Gambau et al., 2016).

The NS algorithm operates in three steps. First, the TB image is oversampled to better identify the nodal points—those where contamination is minimized. Next, the algorithm searches the subpixels of the oversampled image where the Laplacian is minimal, providing an initial estimate of the nodal points. Finally, the search is iteratively refined by identifying subpixels in the oversampled image that minimize the Laplacian of the TB in the original grid. Detailed technical information about the NS methodology can be found in González-Gambau et al. (2015) and González-Gambau et al. (2016).

In González-Gambau et al. (2018), we identified residual contamination near the coast, which was caused by the method used to select the nodal points. The refinement of the initial estimate of the nodal points involved searching for those that minimize the Laplacian of the TB image in the original grid as defined in Eq. (8) of González-Gambau et al. (2015). For clarity in explaining the algorithm's modification, we reproduce the equation below.

$$\bar{t}_i(m,n) = [t_i(m+1,n) + t_i(m-1,n) + t_i(m,n+1) + t_i(m,n-1) + t_i(m+1,n-1) + t_i(m-1,n+1)]/6 \qquad (1)$$

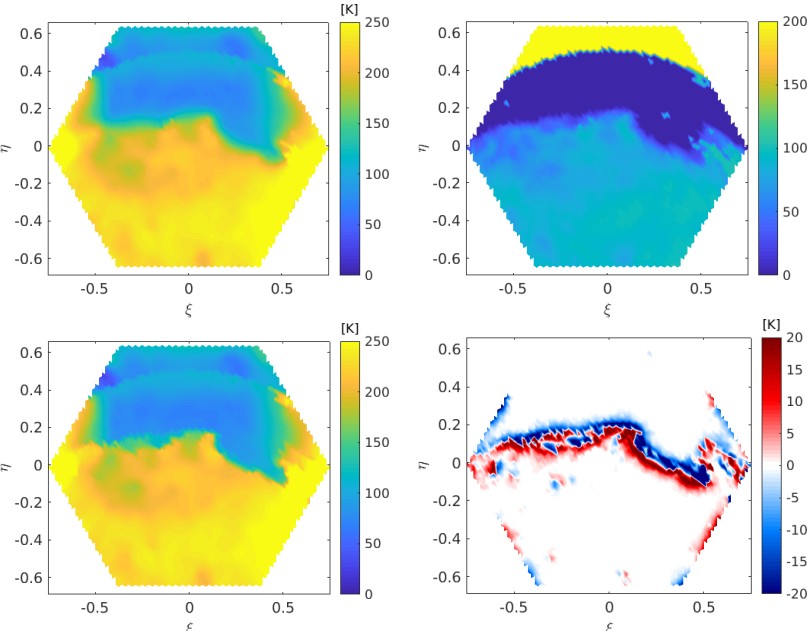

**Figure 2.** Top left: TB image considering the land-sea-sky mask. Top right: land-sea-ice-sky mask. Bottom left: TB considering the land-sea-ice-sky mask. Bottom right: difference between including the sea-ice in the mask and not including it.

To compute the Laplacian at a given point, the average TB is computed considering its six neighboring pixels. However, when land and ocean pixels are mixed, an artificial increase in the ocean TB occurs. To address this, we introduced a land-sea-sky mask when calculating the minimum Laplacian of the TB in the original grid. This modification was crucial to retrieve SSS in semi-enclosed seas (Olmedo et al., 2021b; Grégoire et al., 2023).

Similar to the residual contamination near the coast, an issue arises near ice edges when working in the SO. To mitigate this undesired effect, we introduce in the NS algorithm for the first time a daily sea-ice mask in addition to the land-sea-sky mask. For the construction of the mask, we use a land-sea mask and the SIC values detailed in section 2.1. We interpolate the coast and SIC values to the longitude-latitude coordinates corresponding to each point in the antenna. The mask is equal to 0 for ocean pixels, from 0 to 100 for sea ice pixels with their corresponding SIC value, 100 for land pixels and 200 for sky pixels. Once the land-sea-ice-sky mask is built, the change in the NS algorithm is introduced in Eq. (1) . In the proposed modification, instead of using the 6 closest neighbors for computing the Laplacian, only neighboring pixels with a difference between its mask value and the mask value of the central pixel lower than 10 are considered, meaning that in case of sea-ice pixels, we only consider pixels with the same SIC value $\pm 10\%$. This threshold was selected based on the uncertainty of the SIC product reported in Kreiner et al. (2022). This updated version of the NS algorithm reduces contamination in ocean brightness temperatures near ice margins, as illustrated in Fig. 2. The effect of adding the sea-ice mask is clearly visible. Notably, the differences are concentrated around the sea-ice edge, with lower TB values over the ocean and higher TB values over the ice in the image where the land-sea-ice-sky mask is applied, as expected due to the reduced contamination.

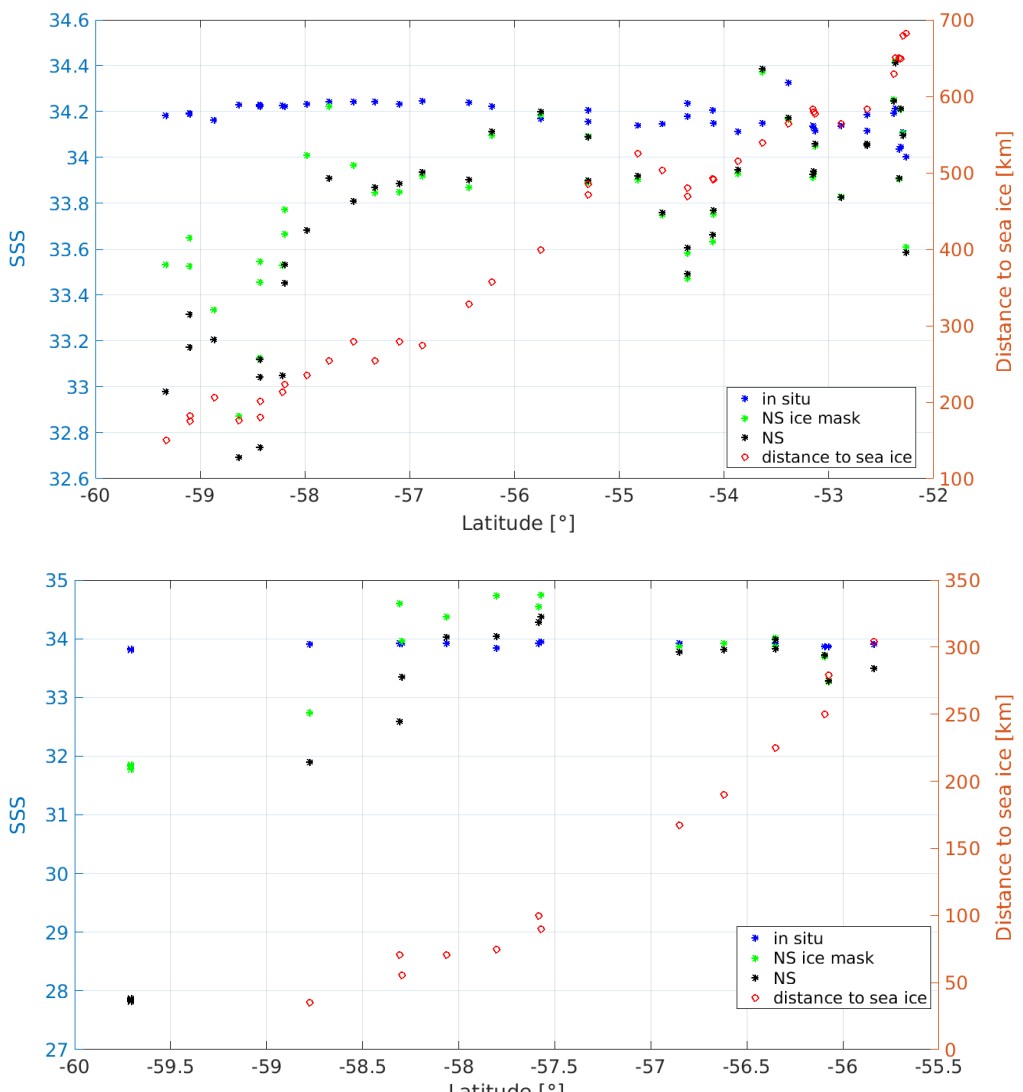

**Figure 3.** Salinity from marine mammals (blue), NS SSS (black) and NS SSS when the ice-sea mask is applied (green) for two different transects. Distance from the sea ice is shown with red diamonds.

We also analyze the impact of including the dynamic sea-ice mask in the retrieved SSS maps by comparing them to in situ measurements. A significant reduction of the artificial freshening close to ice edges can be observed when applying the daily land-sea-ice-sky mask (see transects of Fig. 3). It must be noticed that the SSS values presented throughout the paper refer to Practical Salinity, expressed on the Practical Salinity Scale of 1978 (PSS-78), which is a unitless quantity. Differences between green (NS with land-sea-ice-sky mask) and black (NS with land-sea-sky mask) dots are very small 200 km away from the ice edge, as expected.

### 2.2.3 Reduction of SSS systematic errors : Computation of SMOS climatologies depending on sea-ice distance

In the original formulation of the debiased non-Bayesian (referred hereafter as standard DNB, (Olmedo et al., 2017, 2021a)), raw SSS values are grouped based on across-track distance to the center of the swath ($x$, 50-km bins), incidence angle ($\theta$, 5° bins), geolocation (longitude $\varphi$ and latitude $\lambda$ coordinates) and overpass direction ($d$, ascending and descending), as these factors influence SSS systematic biases. It is considered that systematic errors for all the raw SSS, $s_n^{raw}$, acquired under fixed conditions in the 5-tuple $\gamma = (\varphi, \lambda, d, x, \theta)$ are the same. Then, an estimator of the typical SSS value in the set of measurements $\{s_n^{raw}(\gamma)\}$ can be defined. This typical SSS value is the SMOS-based climatology, which can be used for correcting all the retrievals in the ensemble $\{s_n^{raw}(\gamma)\}$.

During the development of SSS products over semi-enclosed seas, we found that biases also depended on SST (González-Gambau et al., 2022). Consequently, SST was incorporated as an additional variable ($T_s$) in the classification of SSS values, being a 6-tuple $\gamma = (\varphi, \lambda, d, x, \theta, T_s)$ (referred to as the DNB-SST method).

To determine the best strategy for computing the SMOS-based climatology in the SO, we compare the SSS maps generated with the two methods: the standard DNB and the DNB-SST. Twelve years of SMOS SSS retrievals (2011-2022) are used. The SMOS-based climatologies are computed in a rectangular grid of $0.25°$. For each grid cell, the raw SSS values of the eight neighboring cells under the same acquisition conditions $\gamma$ are also included in the distribution to increase the sample size. To minimize the impact of outliers on the statistics, only raw SSS values within the interval between the 5th and 95th quantiles of the distribution are used. For each acquisition condition, the SMOS-based climatology is computed as the mean value of the distribution in the interval $[m_0 - \sigma_0, m_0 + \sigma_0]$ , where $m_0$ corresponds to the mean value and $\sigma_0$ to the standard deviation in the interval [IQ5, IQ95]. This filtering has been selected as a balance between minimizing noise and artifacts while preserving the geophysical signal (García-Espriu et al., 2025). A SMOS-based climatology is discarded if the distribution contains less than 100 measurements or if its standard deviation ($\sigma_0$) exceeds 35. If a given SMOS-based climatology is discarded, all the corresponding $\{s_n^{raw}(\gamma)\}$ are also filtered out, so no SSS will be produced for acquisitions at that geographical location and position in the antenna . It must be noticed that a final value of SSS for that geographical location can be obtained thanks to the contributions of other points in the antenna. Additionally, any raw SSS values that deviate too much from the reference (outside the interval $[m_0 - \sigma_0, m_0 + \sigma_0]$) are also discarded.

SSS maps generated using the standard DNB and DNB-SST methods (see Section 2.2.4) are compared to Astrolabe in situ data (described in Section 3.1.1). The comparison is conducted for 5° latitude bins and across different months to assess the impact of seasonal ice dynamics. The statistical differences between SMOS SSS and in situ measurements are summarized in Table 1. No overall improvement is observed when using the DNB-SST method compared to the standard DNB approach (see the first two rows per each month). In fact, performance degrades between October and December.

To address this, a new version of the DNB, referred to as DNB-ice, is developed specifically for the SO. This version incorporates the distance from sea ice as a variable in the SSS classification acknowledging that sea-ice contamination varies with proximity to the ice edge. In DNB-ice, the fixed conditions are defined as a 6-tuple $\gamma = (\varphi, \lambda, d, x, \theta, Ice)$, replacing the previous use of SST, which was employed in earlier algorithms for regional products (e.g., the Baltic Sea) , by the distance

| | | Mean Difference | | | | STD Difference | | | | N. matchups | | | |
|---|---|---|---|---|---|---|---|---|---|---|---|---|---|
| Month | Dataset | -52 | -57 | -62 | -67 | -52 | -57 | -62 | -67 | -52 | -57 | -62 | -67 |
| January | DNB-SST | -0.08 | -0.04 | 0.23 | 0.02 | 0.19 | 0.12 | 0.40 | 0.92 | 127 | 119 | 131 | 18 |
| | DNB | -0.05 | -0.04 | 0.2 | 0.02 | 0.19 | 0.10 | 0.38 | 0.84 | 127 | 119 | 131 | 17 |
| | DNB-ice | -0.08 | -0.05 | 0.14 | 0.56 | 0.18 | 0.16 | 0.3 | 0.82 | 127 | 119 | 131 | 17 |
| February | DNB-SST | -0.06 | 0.04 | 0.22 | -0.36 | 0.15 | 0.14 | 0.28 | 0.79 | 87 | 92 | 95 | 13 |
| | DNB | -0.03 | 0.04 | 0.12 | -0.27 | 0.14 | 0.1 | 0.21 | 0.72 | 87 | 92 | 95 | 13 |
| | DNB-ice | -0.06 | 0.05 | 0.07 | 0.27 | 0.16 | 0.12 | 0.22 | 0.81 | 87 | 92 | 95 | 13 |
| March | DNB-SST | -0.02 | -0.06 | 0.14 | -1.28 | 0.17 | 0.12 | 0.25 | 0.84 | 119 | 106 | 105 | 7 |
| | DNB | -0.01 | -0.08 | 0 | -1.06 | 0.15 | 0.11 | 0.19 | 0.8 | 119 | 106 | 105 | 7 |
| | DNB-ice | -0.01 | -0.02 | 0.01 | -0.77 | 0.17 | 0.11 | 0.18 | 0.78 | 119 | 106 | 105 | 7 |
| October | DNB-SST | -0.25 | 0.02 | -0.78 | – | 0.17 | 0.18 | 0.56 | – | 36 | 30 | 5 | 0 |
| | DNB | -0.25 | -0.04 | -1.81 | – | 0.2 | 0.27 | 1.54 | – | 36 | 30 | 5 | 0 |
| | DNB-ice | -0.23 | -0.11 | -1.81 | – | 0.18 | 0.26 | 1.39 | – | 36 | 30 | 5 | 0 |
| November | DNB-SST | 0.02 | -0.07 | -0.6 | – | 0.15 | 0.12 | 0.62 | – | 29 | 32 | 13 | 0 |
| | DNB | 0 | -0.05 | -0.69 | – | 0.17 | 0.13 | 0.99 | – | 29 | 32 | 13 | 0 |
| | DNB-ice | 0 | -0.12 | -0.39 | – | 0.15 | 0.1 | 0.61 | – | 29 | 32 | 13 | 0 |
| December | DNB-SST | 0.02 | 0.05 | 0 | – | 0.11 | 0.15 | 0.32 | – | 28 | 30 | 24 | 0 |
| | DNB | 0 | 0.06 | 0.17 | – | 0.12 | 0.15 | 0.58 | – | 28 | 30 | 24 | 0 |
| | DNB-ice | 0.02 | 0.06 | 0.05 | – | 0.11 | 0.14 | 0.21 | – | 28 | 30 | 24 | 0 |

**Table 1.** Monthly statistics of the difference between SMOS SSS and Astrolabe in situ measurements, grouped by latitude bins and categorized by the three different versions of the DNB.

from sea ice edge. The SIC values defined in section 2.1 are employed to calculate the distance from sea ice, with five bins considered: the first 100 km, [100-200 km], [200-300 km], [300-400 km], and beyond 400 km. An example of the distance from sea-ice is shown in Fig. 4. The mean and standard deviation of the SSS distributions are shown in Fig. 5 for the first bin (the first 100 km from sea-ice edge) on the top panel and for the fifth bin (beyond the first 400 km) on the bottom panel. As
expected, higher biases and standard deviations are obtained for the first 100 km from the sea ice edge. Salinity maps generated with DNB-ice are also compared to Astrolabe in situ data. Statistics are presented in the third row of each month in Table 1. Overall, both the mean and the standard deviation of the difference between satellite and in situ salinities are reduced near the ice edge when distance-to-sea-ice bins are used, compared to the standard DNB and DNB-SST methods. Consequently, the DNB-ice approach is used for generating the BEC SO SSS product.

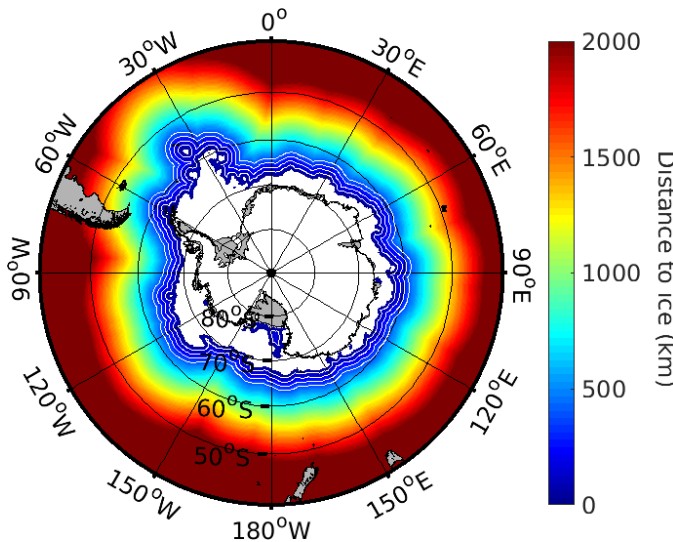

**Figure 4.** Distance from sea ice [km] of a given date (2021-01-01). Five bins of distance from the sea ice (white contours) are considered in the computation of the SMOS-climatologies.

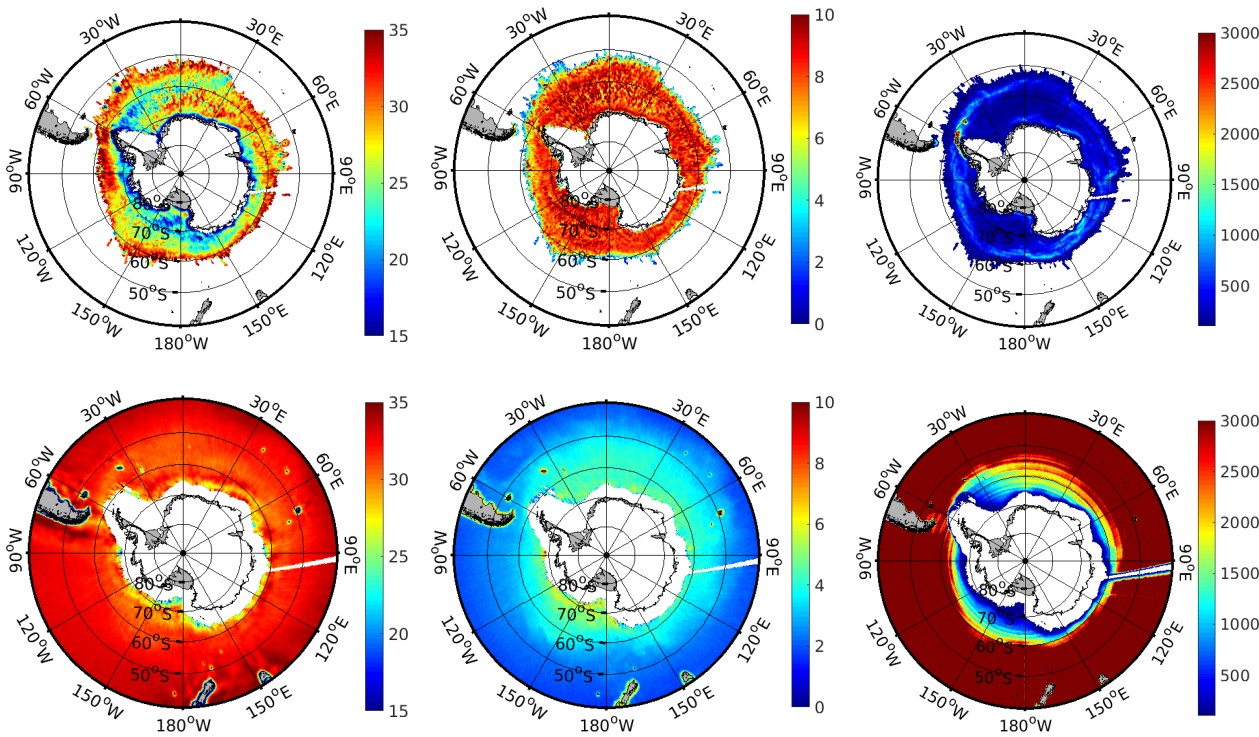

**Figure 5.** Statistics of SSS distributions for ascending overpasses, at the center of the swath ($x = 0$) and incidence angle $\theta = 42.5°$. Top: for the first 100 km (bin 1), bottom: for distances beyond 400 km (bin 5). Left: SMOS-based climatology, middle: standard deviation of SSS distribution and right: number of measurements.

### 2.2.4 Generation of the BEC SO SSS product

**Generation of debiased SSS salinities**

Debiased absolute salinity values are obtained by adding the multiyear SSS reference to the debiased non-Bayesian SSS anomalies. The annual WOA13 SSS climatology is used for this reference. We estimate the uncertainty for each individual SSS by propagating the radiometric errors on the TB to the SSS: $\epsilon = 0.5(\sigma_H + \sigma_V)/\Delta T'$, where the $\sigma_H$ and $\sigma_V$ are the radiometric sensitivities for H and V polarizations, respectively and the term $\Delta T'$ is a numerical estimate of the derivative of the TB with respect to SSS (see more details in Olmedo et al. (2021a)).

Each Level 3 map is produced by calculating a weighted average of the debiased and filtered SSS values for a given grid point across all overpasses during the 9-day period, with weights inversely proportional to the squared uncertainty.

**Mitigation of temporal biases** Since the SMOS-based climatologies integrate data over a multi-year period, these do not change along time (Olmedo et al., 2017). Therefore, a strategy for correcting the temporal biases need to be introduced at this point of the processing.

Initially, we attempted to correct the maps using the temporal correction applied to BEC global SSS maps (Olmedo et al., 2017, 2021a) by assuming that the global SSS average is constant over time. In this way, we avoid the use of in situ measurements as a reference, maintaining as much as possible the surface dynamics that are measured by the satellite in the first centimeters of the upper ocean layer (Olmedo et al., 2022). To apply this temporal correction, we used global maps generated with the same methods employed for the BEC SO SSS product. However, after applying this temporal correction, seasonal discrepancies with respect to Argo measurements were observed. Consequently, the final approach consists of computing the temporal correction as the mean difference between the 9-day satellite SSS map and the collocated Argo salinity over the same period. Specifically, a single value is subtracted from each 9-day SSS map. This temporal correction was also applied in the development of the BEC Arctic SSS former v2.0 product (Olmedo et al., 2018). Since Argo floats are used in calculating the temporal correction, they are excluded from the validation process.

**Filtering of SSS maps**

To enhance the quality of the SSS maps, an additional filtering criterion is applied in the final processing step. All the SSS values greater than 40 and/or with uncertainties exceeding 10 are filtered out. This criterion is based on the 2D histograms for each season relating SSS values to their associated uncertainties and to the distance from sea ice (see Fig. 6). Higher uncertainty values are found in periods of sea ice melting (October-March). Notably, the filtered data are primarily concentrated in the first bin closest to the ice edge, i.e., within the first 50 km.

**BEC SO SSS product v1.0**

The BEC SO SSS product v1.0 provides 9-day SSS maps at a 25 km EASE-SL grid (Brodzik et al., 2012), generated daily (González-Gambau et al., 2023). The time series covers the period from February 1, 2011, to March 31, 2023. The product spans latitudes south of 30°S and is distributed in netCDF files, which include both the SSS values and their associated uncertainties.

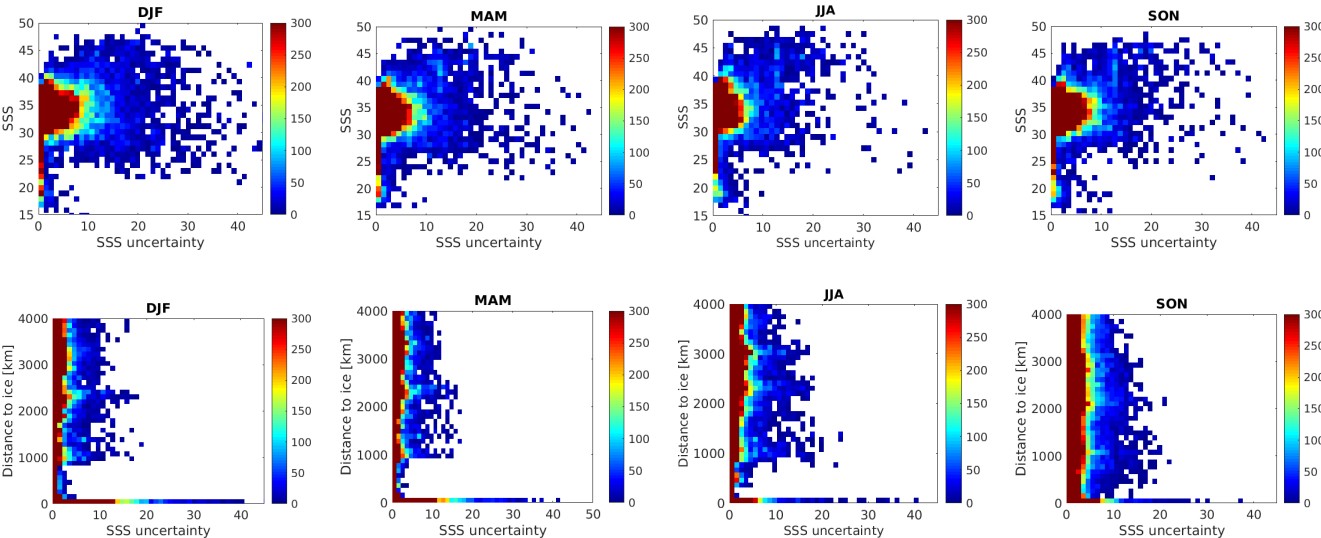

**Figure 6.** Top row: 2D histograms of SSS with respect to its uncertainty for each season. Bottom row: 2D histograms of distance from sea ice with respect to uncertainty for each season.

Two examples of BEC SO SSS maps are shown in Figure 7. The left panel displays the SSS values, while the right panel shows the corresponding uncertainties. The maps at the top correspond to February 15, 2020, representing the period of minimum sea ice extent in the year, while the maps at the bottom correspond to September 15, 2020, representing the period of maximum sea ice extent in the year. Additionally, the reopening of the Weddell Polynya during the winter of 2017 is illustrated in Fig. 8, marking its return since the 1970s.

## 3   Quality assessment of the BEC SO SSS product

### 3.1   Datasets for inter-comparison and validation

The quality assessment of the BEC SO SSS product results from the comparison against the reference in situ datasets and ocean models presented in this section.

#### 3.1.1   In situ salinity measurements

- *Thermosalinograph (TSG) data by the Università degli Studi di Napoli Parthenope*: Data collected in the Atlantic (Aulicino et al., 2018a, b) and Pacific sectors by Italian and South African icebreakers. Data are collected along the routes from South Africa to Antarctica and from New Zealand to Antarctica. Ships tracks cross the Antarctic Circumpolar Current (ACC) and its fronts during the austral summer, in the framework of the yearly Antarctic expedition of the Southern National Antarctic Program and of the Italian Programme for research in Antarctica (Programma Nazionale di Ricerche

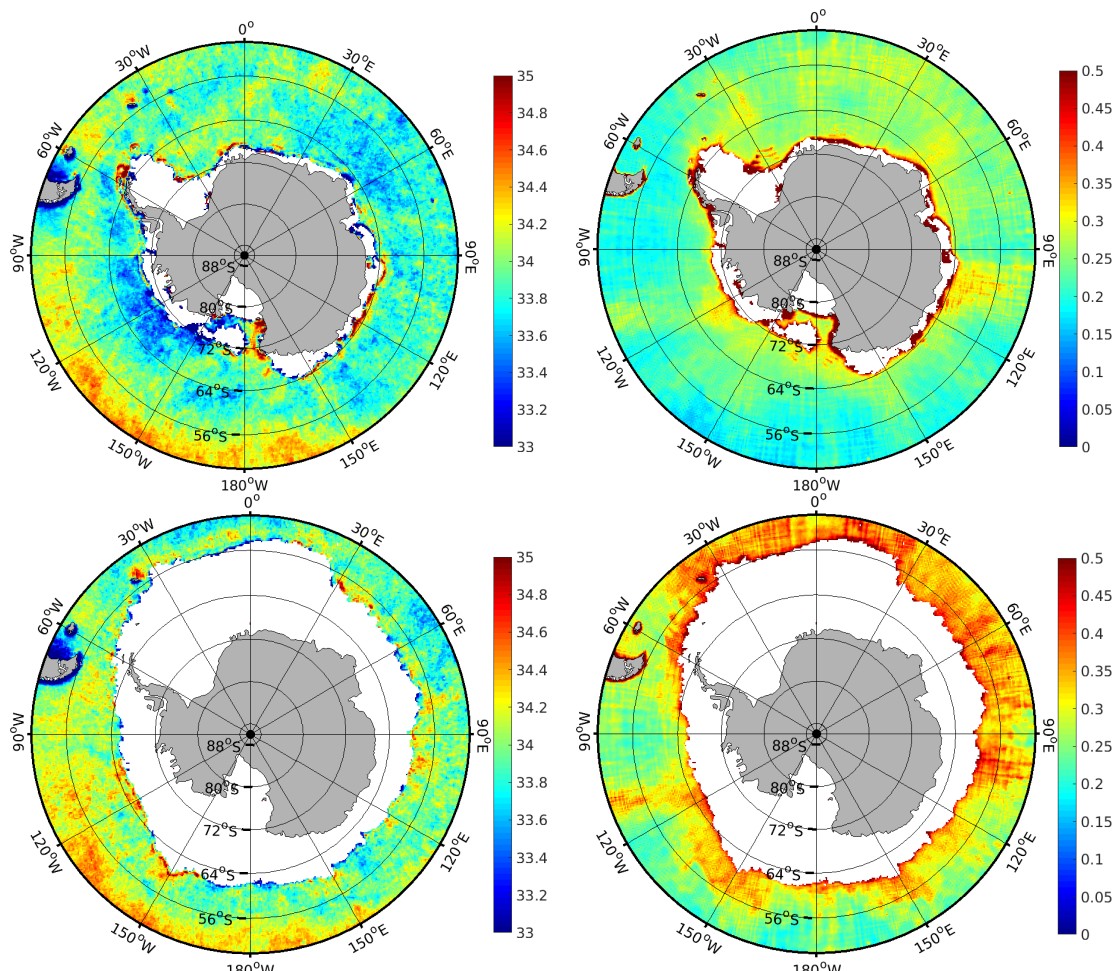

**Figure 7.** Top: 9-day BEC SO SSS map as of February 15, 2020 (left) and its uncertainty (right). Bottom: 9-day BEC SO SSS map as of September 15, 2020 (left) and its uncertainty (right).

in Antartide-PNRA). The depth of these measurements is approximately 5 m. These datasets are particularly valuable for analyzing the quality of the satellite SSS product, both at mid-latitudes and in proximity to sea ice.

- *TSG data by the Astrolabe vessel*: TSG salinity measurements provided by the observing ships network in the West Pacific Ocean (Morrow and Kestenare, 2014). This dataset is provided by the Survostral project and is available at https://www.legos.omp.eu/survostral/data-products/tsg-sss-sst/. These transects routinely cross the ACC from Tasmania to the Antarctic Continent. Measurements from 2014, 2015, 2020, and 2022 are used. When available, only adjusted values and with good quality (quality_flag=1) are considered. The depth of these measurements in between 5-10 m. This
dataset is valuable for assessing seasonal and latitudinal biases, which in this region are strongly influenced by sea-ice contamination.

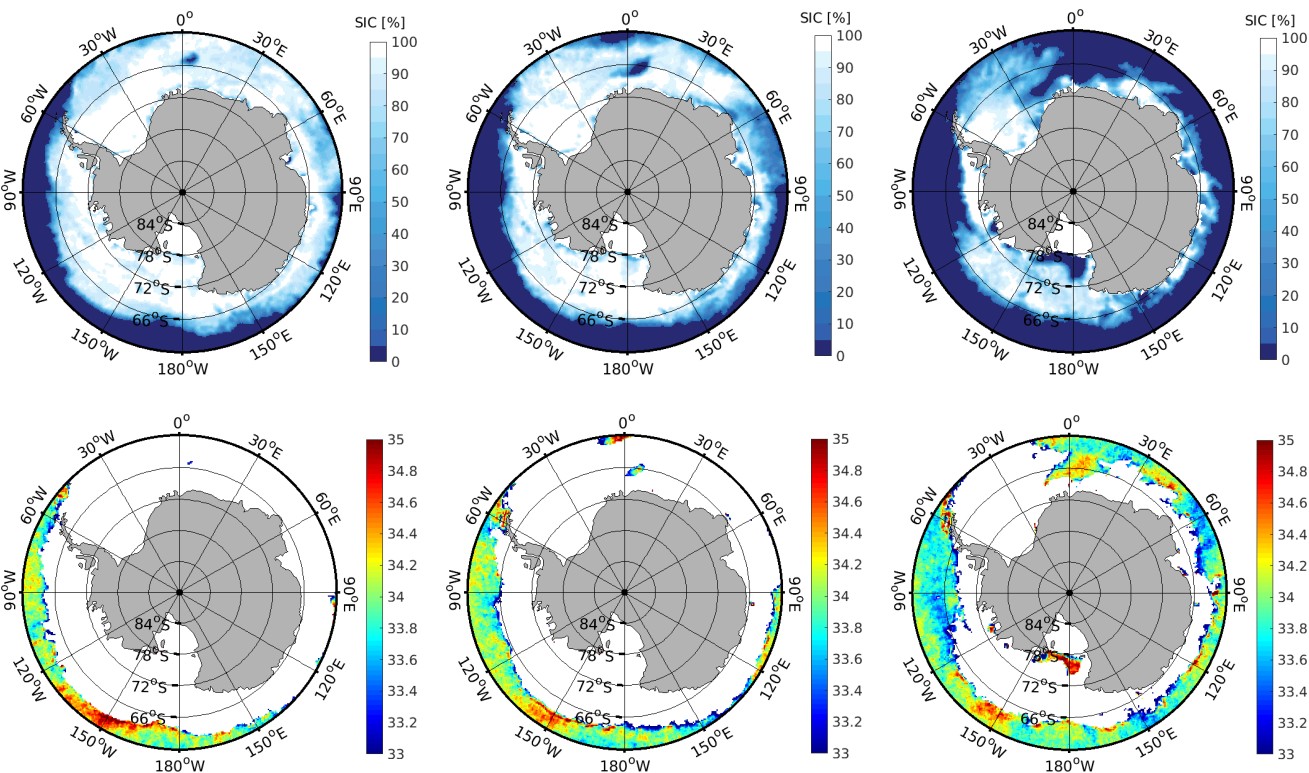

**Figure 8.** Top: SIC percentage, bottom: BEC SO SSS maps for the following dates: 15 October 2017, 15 November 2017 and 15 December 2017. Note that the polynya is not seen in the SSS maps until there is no sea ice (SIC=0) because of the filtering we apply in the raw SSS.

- *Marine mammals*: The marine mammal dataset (MEOP-CTD database, 2024-02 release, (Roquet et al., 2024) with an accuracy of $\pm0.05$ (Treasure et al., 2017) has been used. This dataset was collected and made available by the International MEOP Consortium (http://www.meop.net) and the national programs that contribute to it. Measurements
used in the validation correspond to the shallowest profile (in the first 10 m, with the most usual depths lower than 4 m) and only those measurements of good quality (quality control equal to 1). This dataset includes extensive data during winter, a period typically not covered by ships.

- *Barcelona World Races (BWR) 2011 and 2015 (Umbert et al., 2023) and 2020 Vendée Globe (DOFT-ICM, 2023; Umbert et al., 2022)*: South of $40°S$, these routes run almost parallel to the ACC going through the Southern ACC Front, the
Polar Front, and the Sub-Antarctic Front in some places. These measurements provide surface information at 60 cm depth, which is not available from other in situ sensors. Details on the filtering applied to these datasets can be found in (Salat et al., 2013), (Umbert et al., 2022) , (Hernani et al., 2025) . These datasets enable the quality assessment of the product in the Sub-Antarctic zone. (Hernani et al., 2025)

- *Global Ocean-Gridded objective analysis fields of salinity (Product ID: INSITU_GLO_PHY_TS_OA_MY_013_052)* are generated using profiles from CORA 5.2 (INSITU_GLO_TS_REP_OBSERVATIONS_013_001_b) (Szekely et al., 2019, 2024). Monthly maps of the global ocean at $0.5°$ resolution, focusing on the shallowest level (1 m), are used to compare interannual and seasonal salinity variability.

### 3.1.2   Ocean models

- *The GLORYS12V1 product* is a global ocean eddy-resolving model provided by CMEMS. The global ocean output files are in a regular grid of $1/12°$ and includes 50 vertical levels. This product is available at  https://data.marine.copernicus.eu/product/GLOBAL_MULTIYEAR_PHY_001_030/files?subdataset=cmems_mod_glo_phy_my_0.083deg_P1M-m_202311 (Mercator Océan International, 2023). We use the monthly mean salinity fields provided at 0.5 m depth, which are re-gridded to the satellite SSS grid.

- *The Biogeochemical Southern Ocean State Estimate (B-SOSE)* is a numerical ocean model based on the MITgcm, with 52 vertical layers that are more closely spaced near the surface, and a horizontal grid of $1/6°$ that is also eddy-permitting (Verdy and Mazloff, 2017). B-SOSE assimilates observations from Lagrangian floats, remotely sensed sea surface height, sea surface temperature, and sea ice concentrations using a 4D-Var technique that does not break con-servation of physically conservative quantities (Mazloff et al., 2010). We use the Iteration 135 (available at https://sose.ucsd.edu/SO6/ITER135/) and the salinity value of the uppermost layer. The model data is provided as a 5-day average every 5 days. The common period between the model and satellite data is from 2013 to 2018. The model data is re-gridded to the satellite SSS product grid and the temporal collocation has been done for the same central day.

## 3.2   Validation methods and quality metrics

### 3.2.1   Collocation strategy

When in situ measurements are available at different depth levels, the shallowest measurement is used for validation. The locations of in situ data are assigned to the nearest satellite grid cell. For in situ measurements acquired with high temporal frequency (such as the TSG data), all the measurements which fall in the same satellite grid cell are averaged. In terms of temporal collocation, all the in situ available in the 9 days used to generate the satellite SSS product are considered in the validation of the corresponding satellite SSS map.

### 3.2.2   Quality metrics

The validation metrics are based on the statistics of the difference between the satellite SSS and in situ salinity ($\Delta SSS = SSS_{sat} - SSS_{insitu}$) at the various matchups. The following metrics are computed:

- Global statistics of $\Delta SSS$ for each in situ dataset, calculated annually.

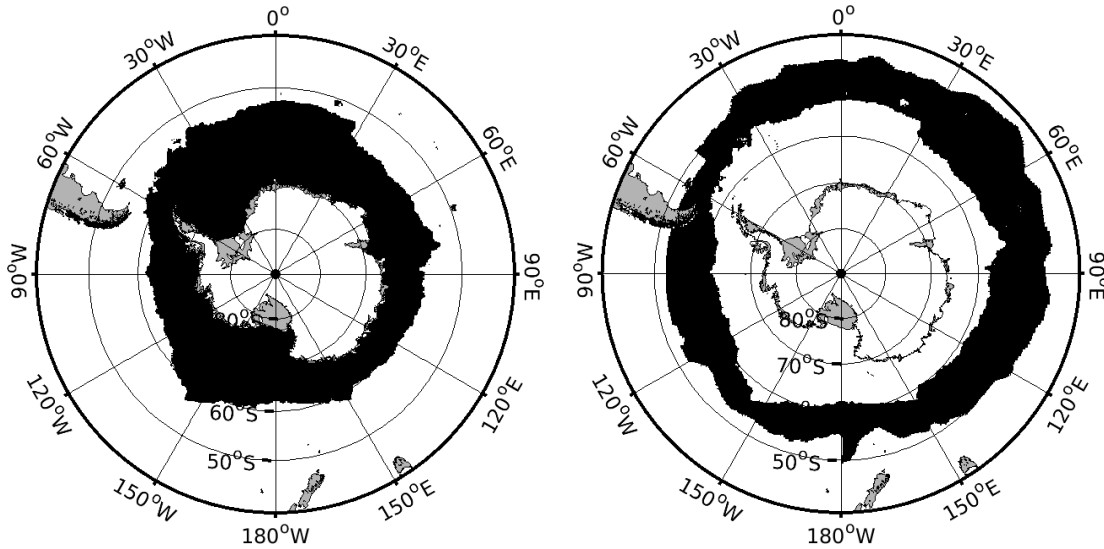

**Figure 9.** Left: Subpolar region. Right: subAntarctic region.

- Maps showing the spatial distribution of $\Delta SSS$ statistics, including the temporal mean of $\Delta SSS$, the temporal standard deviation of $\Delta SSS$, and the number of matchups for each grid cell. All $\Delta SSS$ values at the collocated points are aggregated into a $1° \times 1°$ grid.

- 2D histograms of $\Delta SSS$ statistics as a function of distance from the ice edge (x-axis) and distance from the coast (y-axis), binned in 50 km intervals. This metric is particularly useful for identifying residual land-sea and ice-sea contamination.

For the evaluation of satellite SSS dynamics, we compute:

- Time series of monthly average SSS in the Subpolar and Sub-Antarctic regions. The Subpolar region spans from the coastline to the maximum sea ice extent in the analyzed period. The Sub-Antarctic region is defined between the 34.1 isohaline and the line of maximum sea ice extent (see defined regions in Fig. 9). We compare the time series of satellite-derived monthly average SSS with those from CORA, GLORYS, and SOSE. For this comparison, all the datasets are filtered based on the satellite coverage.

- Seasonal and interannual variability of satellite SSS product and regional model: First, the model output is regridded to the satellite grid and filtered based on satellite coverage. Then, the interannual variability is computed as the annual average SSS with respect to the average of the period when the model is available (2013-2018). The seasonal variability is computed as the seasonal average SSS (DJF, MAM, JJA, SON) with respect to the complete period.

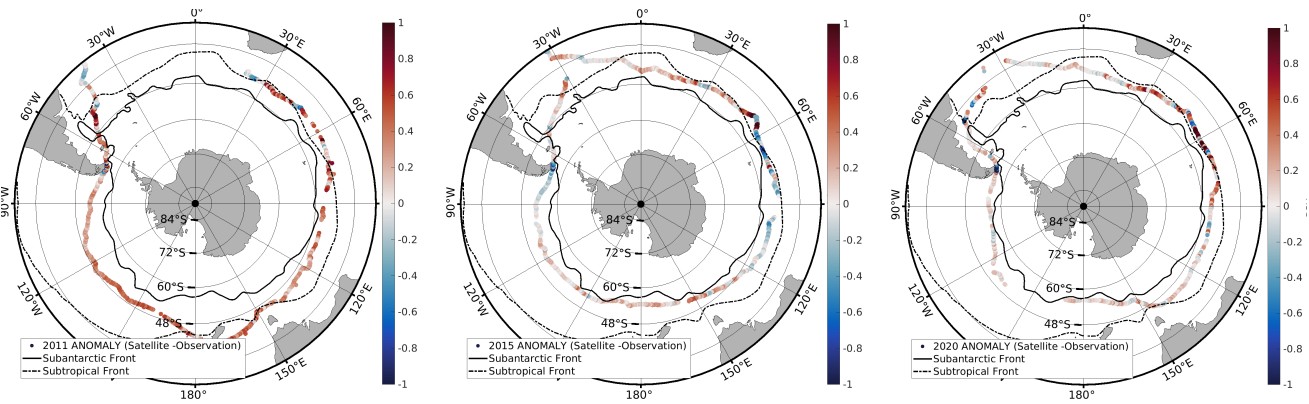

**Figure 10.** Mean difference between satellite SSS and in situ measurements from Barcelona World and Vendee Globe races for 2011, 2015 and 2020 (from left to right).

## 3.3   Validation results

### 3.3.1   Performance in the sub-Antarctic region

We use the in situ data from the Barcelona World Races and Vendée Globe Race for analyzing the performances in the sub-Antarctic region. The analysis is performed for each of the three years separately. Statistics of the differences between satellite SSS and in situ measurements are collected in Table 2. We observe a strong correlation between the two datasets, as evidenced by the high correlation coefficient. A map of the difference between satellite SSS and in situ measurements is shown in Fig. 10.

In Fig. 11, the salinity values of satellite and in situ and the difference between both are shown as a function of the longitude. A good correspondence between both datasets is observed. The highest biases are found for the year 2011. This also happens when comparing satellite data to the WOA climatology (not shown). Larger differences are observed when changes in salinity values occur more rapidly. These differences are not only attributable to satellite measurement errors but also to the different spatial and temporal scales resolved by the satellite and in situ sensors (an integrated area over 9-day versus punctual and instantaneous measurements).

| Dataset | Time | Bias | Std.dev. | R | Slope | Y-Intercep | N.matchups |
|---------|------|------|----------|------|-------|------------|------------|
| BWR | 2011 | 0.3 | 0.33 | 0.83 | 0.78 | 7.88 | 1187 |
| BWR | 2015 | 0.05 | 0.27 | 0.78 | 0.7 | 10.4 | 1601 |
| VG | 2020 | 0.1 | 0.31 | 0.81 | 0.83 | 5.89 | 1337 |

**Table 2.** Statistics of the differences between satellite and in situ measurements for each year.

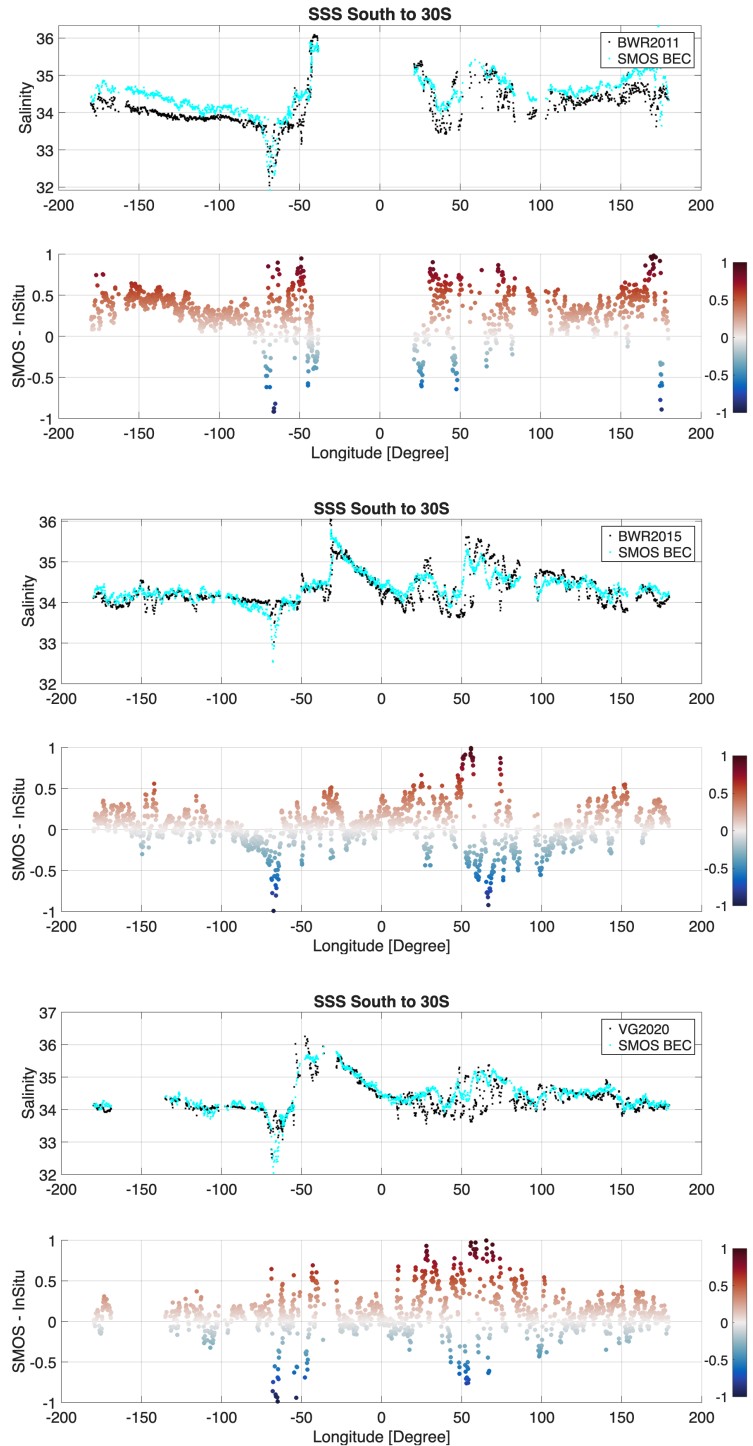

**Figure 11.** BEC SO SSS, in situ salinity and the difference between both as a function of the longitude from Barcelona World Races and Vendee Globe Race for the years 2011, 2015 and 2020 (from top to bottom).

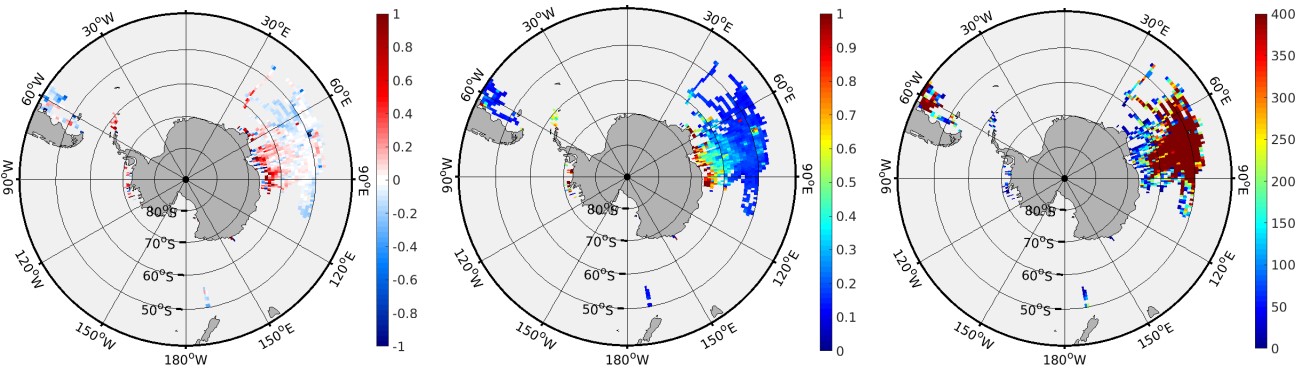

**Figure 12.** Spatial statistics (in a $1°$ grid) of the difference between satellite SSS product and marine mammals measurements. Left: Mean $\Delta SSS$, center: standard deviation of $\Delta SSS$, right: number of matchups.

### 3.3.2 Performance near coasts and ice edges

The performance of the BEC SO SSS product close to coasts and ice edges is primarily assessed using measurements acquired by marine mammals and TSG data along the Atlantic and Pacific sectors of the Southern Ocean provided by the Università

degli Studi di Napoli Parthenope. It is important to note that the two in situ datasets exhibit several sampling differences. First, their spatial coverage differs significantly. TSG data are mostly regularly distributed along the meridional transects followed by the research vessels across various sectors of the SO. Although concentrated on very tailored longitudes, they provide a relatively circumpolar representation of in situ conditions and occasionally extend into more southerly latitudes (e.g., beyond $65°S$ in the Ross Sea) exceeding the spatial extent of the marine mammals dataset. In contrast, the marine mammals dataset

is largely confined to a single sector of the SO, with sampling mainly concentrated near the sea ice edge. Second, there is a marked difference in the temporal distribution of the two datasets. Observations from marine mammals are predominantly acquired during the winter months, whereas TSG data are primarily collected during the summer.

The spatial maps of the temporal mean and the temporal standard deviation of the $\Delta SSS$ and the number of matchups are shown in Fig. 12 and Fig. 14 for the comparison to marine mammals and TSG, respectively. As expected, larger differences

are concentrated very close to coasts and sea ice edges, particularly in the marine mammals dataset.

To better analyze the performance close to coasts and sea ice, we compute the 2D histograms of the mean and the standard deviation of the $\Delta SSS$ as a function of the distance from ice edges and to coasts. Results are shown in Fig. 13 and Fig. 15 for the comparison to marine mammals and TSG, respectively. The statistics reveal almost negligible biases with standard deviations of 0.2 for marine mammals and 0.25 for TSG measurements. Differences in statistics may stem from the previously

mentioned sampling differences. In both cases, the largest errors occur within the first 100-150 km from the sea ice edge and/or the coastline, as shown in Fig. 16.

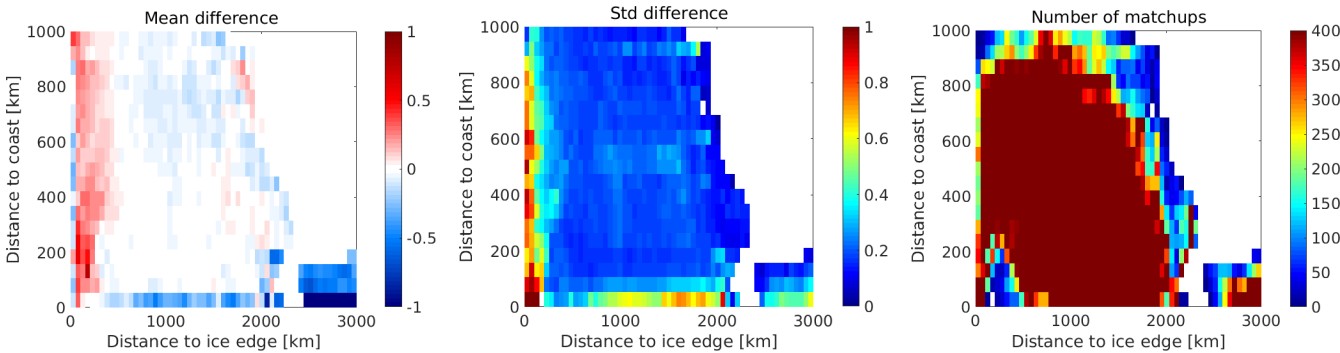

**Figure 13.** 2D histograms of the difference between satellite SSS product and marine mammals measurements as a function of the distance from the ice edge and distance to coast. Left: Mean $\Delta SSS$, center: standard deviation of $\Delta SSS$, right: number of matchups.

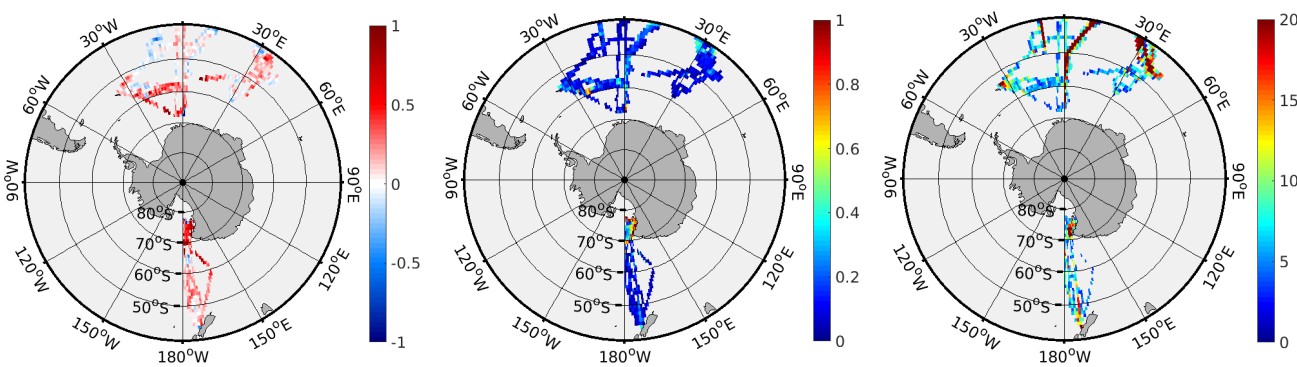

**Figure 14.** Spatial statistics (in a $1°$ grid) of the difference between satellite SSS product and TSG measurements from ships of opportunity. Left: Mean $\Delta SSS$, center: standard deviation of $\Delta SSS$, right: number of matchups.

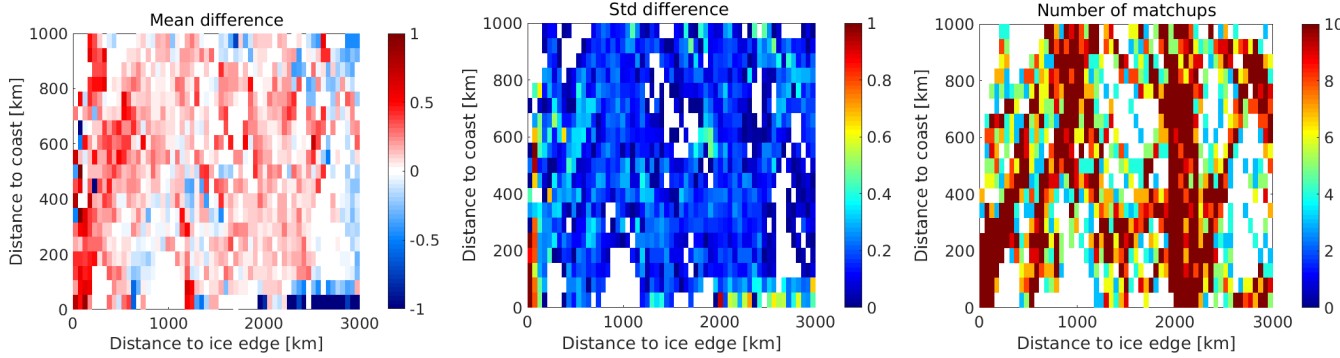

**Figure 15.** 2D histograms of the difference between satellite SSS product and TSG measurements as a function of the distance from the ice edge and distance to coast. Left: Mean $\Delta SSS$, center: standard deviation of $\Delta SSS$, right: number of matchups.

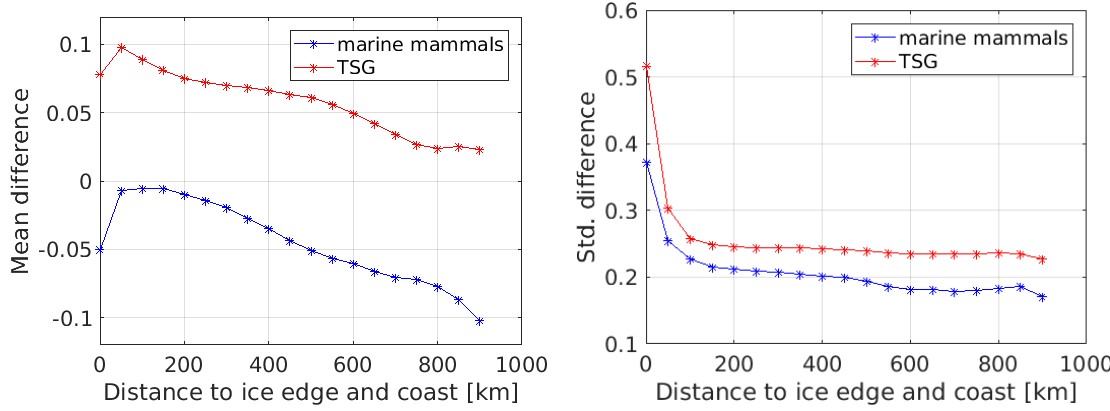

**Figure 16.** Left: Mean $\Delta SSS$, right: standard deviation of the $\Delta SSS$ for marine mammals and TSG for each bin of the distance from sea ice and/or coast.

### 3.3.3 Seasonal and interannual variability

In this section, we analyze the average satellite SSS over two regions: the Subpolar region and the Sub-Antarctic region (Fig. 9. We compare the monthly time series of satellite-derived average SSS with those from the CORA dataset and GLORYS and
380 SOSE models. From this analysis (see Fig. 17 and Table 3), we observe that in the Sub-Antarctic region, all datasets show good agreement, although a mean bias of -0.16 is found between the SOSE model and the other three sources. This is consistent with the differences of -0.2 found with respect to Argo at 7 m depth (https://sose.ucsd.edu/RESEARCH/BSOSE6/ITER135/SALT_ Argo_BSOSE.html). The temporal variability observed across the different salinity values is very similar between BEC SO, CORA and GLORYS, with correlations of 0.7. However, the SOSE model shows a much lower correlation with the BEC SO
product in the Sub-Antarctic region. When analyzing the Subpolar region, the differences between satellite SSS and CORA are larger during periods of higher stratification, likely due to greater differences between satellite-based surface measurements, which capture only the top few centimeters, and in situ measurements that represent deeper layers. Additionally, we observe biases between both models and satellite SSS and CORA. Despite these biases, the temporal variability in satellite SSS is more closely aligned with the SOSE model than with the GLORYS model. This is primarily because GLORYS uses a 3D variational
data assimilation which breaks physical conservation, whereas SOSE utilizes a 4D variational data assimilation. As a result, the physical consistency of the dynamics and thermodynamics in SOSE is maintained by the assimilation scheme, often leading to more accurate representation of the flow (Abernathey et al., 2016; Narayanan et al., 2024).

We also analyze the interannual and seasonal variability of the BEC SO SSS product and compare it to the one shown by the regional SOSE model. Overall, the interannual variability shown by the satellite product (first and third rows in Fig.
18) is consistent with the variability shown by the SOSE model (second and fourth rows), although larger SSS interannual variability is shown by the model. Satellite and model show a freshening in the initial years and a salinification since 2016, which is associated with the sea ice decline in Antarctica (Purich and Doddridge, 2023), (Silvano et al., 2024). The patterns

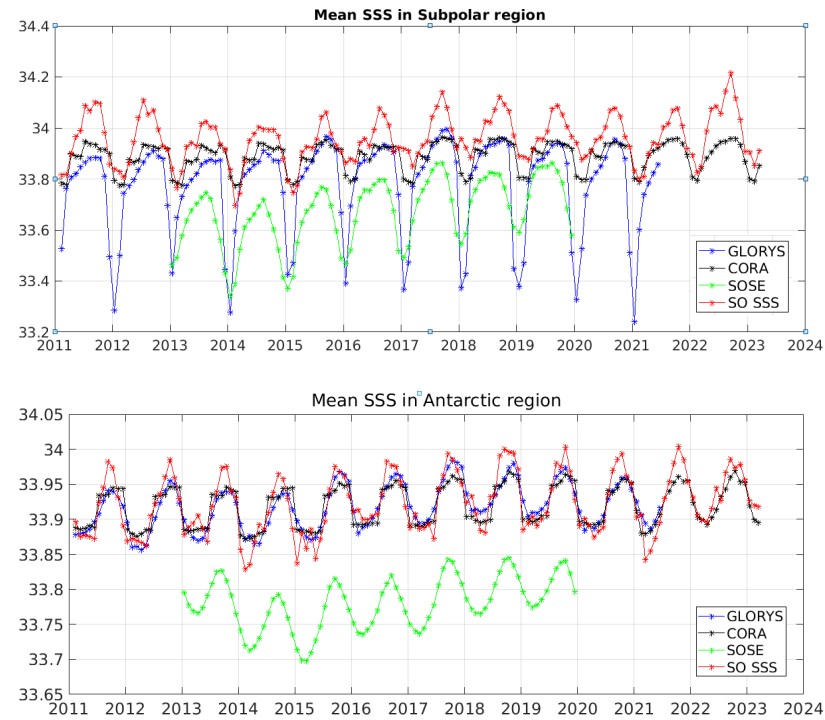

**Figure 17.** Top: temporal series of the monthly average salinity in the Subpolar region. Bottom: temporal series of the monthly average salinity in the subAntarctic region.

| | Antarctic region | | | Subpolar region | | |
|---|---|---|---|---|---|---|
| Datasets | Mean | Std.dev. | R | Mean | Std.dev. | R |
| CORA - BEC SO SSS | -0.01 | 0.02 | 0.69 | -0.06 | 0.05 | 0.83 |
| GLORYS - BEC SO SSS | -0.01 | 0.02 | 0.73 | -0.16 | 0.15 | 0.59 |
| SOSE - BEC SO SSS | -0.16 | 0.04 | 0.24 | -0.28 | 0.09 | 0.75 |

**Table 3.** Statistics of the comparison of the time-series shown in Fig. 17 in the common period 2013-2018.

remain consistent even near the ice edge, demonstrating that, despite larger errors in the satellite product close to the ice, the satellite-derived SSS is capable of capturing the SSS variability in close proximity to the ice edge.

Similar conclusions can be drawn from the analysis of seasonal variability shown by the satellite SSS and by the SOSE model (Fig. 19). Seasonal variability is consistent with the expected processes occurring near Antarctica: lower salinity between November and March (austral spring and summer) when sea ice melts, and progressively higher salinity between April and October (austral autumn and winter) due to the surface brine rejection when sea ice forms.

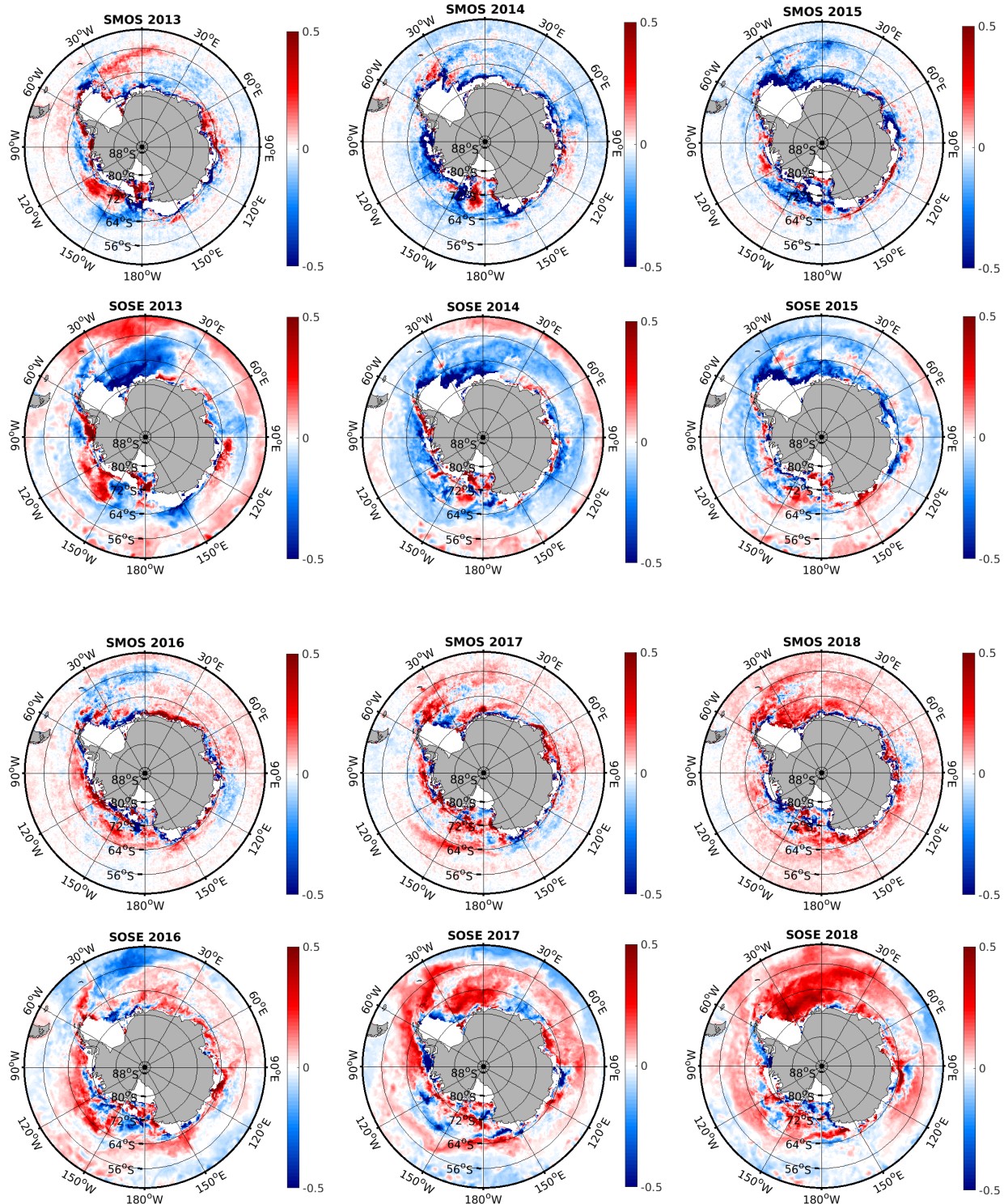

**Figure 18.** Interannual variability of: satellite SSS product for the period 2013-2015 (first row), SOSE salinity for the period 2013-2015 (second row), satellite SSS product for the period 2016-2018 (third row) and SOSE salinity for the period 2016-2018 (fourth row).

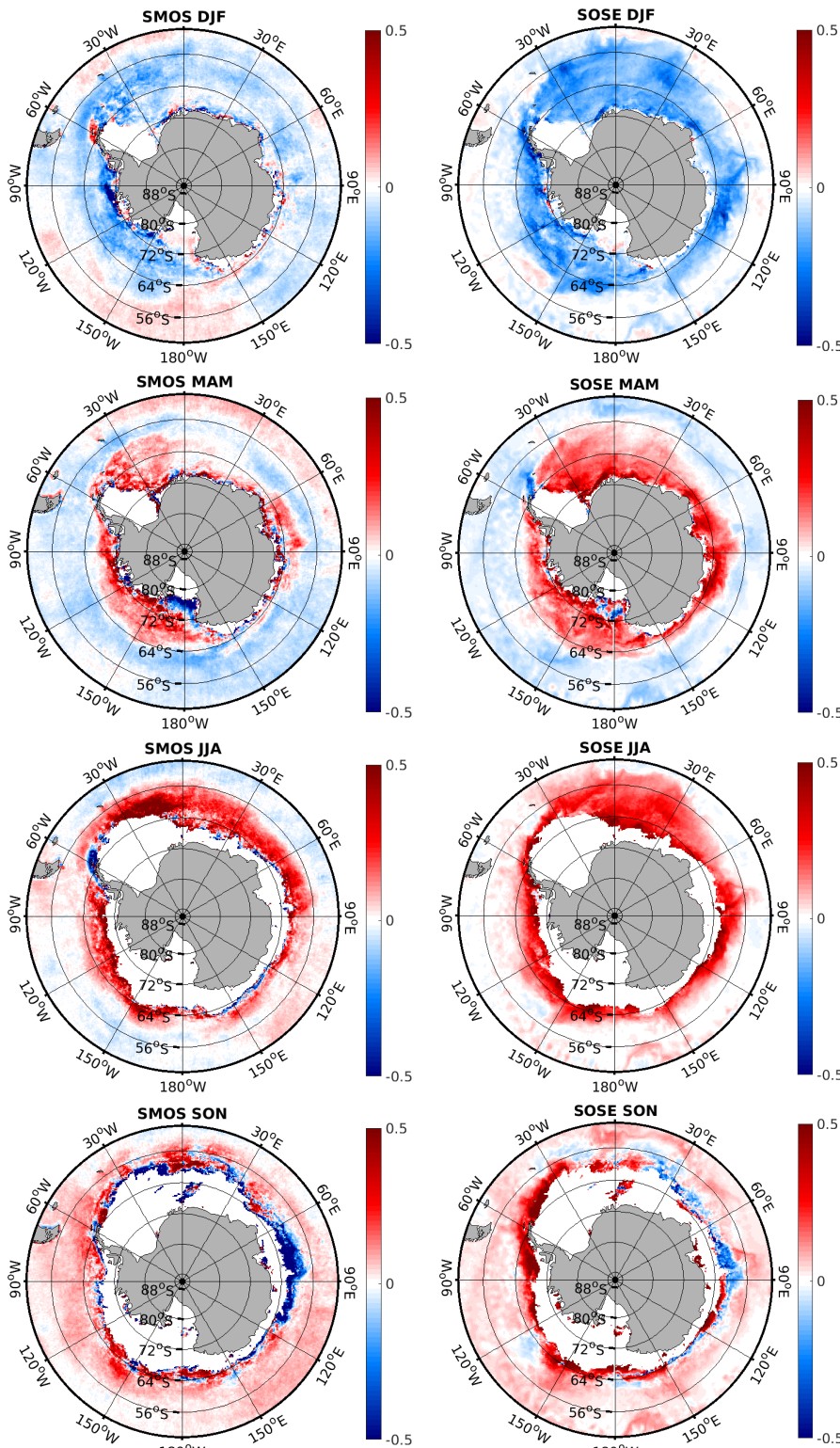

**Figure 19.** Seasonal variability of satellite SSS product (left column) and SOSE salinity (right column) for the period 2013-2018.

## 4 Conclusions

In this paper, we present the dedicated BEC SSS product over the Southern Ocean. The primary algorithms used to retrieve SSS are specifically designed to minimize sea-ice contamination. Notably, for the first time, we employ a dynamic sea-ice mask when applying nodal sampling, which significantly reduces TB radiometric errors near the ice edges. Additionally, we apply the debiased non-Bayesian retrieval scheme to characterize and mitigate spatial biases in SSS, taking into account both the acquisition conditions and, for the first time, the distance from ice edges to minimize sea-ice contamination. These algorithm improvements have also directly contributed to the latest release of Arctic L3 SSS maps (v4), which were recently developed and distributed by BEC (García Espriu et al., 2024) and have a significant impact on the enhancement of the SSS retrievals from satellite L-band measurements more broadly.

The use of various in situ datasets has enabled both global and seasonal analyses, as well as assessments based on distance from ice edges. The BEC SO SSS product demonstrates very high performance beyond 100-150 km from the sea ice edges, with nearly zero bias and a standard deviation of the difference of 0.22 when compared to marine mammal data and 0.25 when compared to TSG data from research vessels. Within the first 100-150 km from the sea ice edges, some in situ measurements indicate good consistency with satellite data in capturing freshwater near the ice. However, satellite measurements sometimes report higher salinity values compared to in situ observations. Some of these differences can be attributed to variations in the spatial and temporal scales captured by the different instruments. In more dynamic regions, such as those close to the ice edges or coasts, sampling-related errors are expected to be more pronounced. Therefore, part of the high variability observed within the first 100-150 km from the coast and ice margin reflects geophysical signals, and is also captured by the regional SOSE model. However, some of this variability could be attributed to residual contamination in the satellite-derived SSS product. Further research is required to understand which part of the variability is still residual contamination and then, to further enhance the quality near the boundaries, enabling more accurate capture of critical processes occurring in close proximity to ice margins.

This new satellite SSS product provides a reliable description of the ACC and captures seasonal and interannual variabilities aligned with those observed in the SOSE regional model. This product has been used in the study by Silvano et al. (2025). Using these satellite observations, a marked increase of SSS across the circumpolar SO has been observed since 2015. This is a reflection of a weakened upper-ocean stratification, coinciding with the notably Antarctic sea ice retreat. Besides, the weakened upper-ocean stratification reflected by the satellite salinity product, also contributed to the reappearance of the Maud Rise polynya in the Weddell Sea, during the winters of 2016-2017 (Campbell et al., 2019). Future work will focus on further evaluating the quality of SSS in the polynyas.

Current models of the Southern Ocean poorly represent several climate-critical processes, such as sea ice melting/freezing, upper-ocean mixing, bottom water production at high latitudes, and the formation of coastal and open-ocean polynyas. All these processes are controlled by upper-ocean salinity (e.g. (Silvano et al., 2023); (Narayanan et al., 2024); (Goosse and Zunz, 2014)). Incorporating these new satellite SSS observations into data-assimilating models (e.g., the SOSE and GLORYS models)

will thus boost their ability to reproduce the observed changes, helping us to understand what key dynamics must be better represented by climate models in order to credibly project the future of the Southern Ocean.

## 5 Data availability

The DOI of the BEC SO SSS product is: https://doi.org/10.20350/digitalCSIC/15493 (González-Gambau et al., 2023). Access to the data is provided by the Barcelona Expert Center, through its FTP service. The product with their metadata and associated documentation can be downloaded using the following credentials:

Host: sftp://eodata-bec.icm.csic.es

Username: ftpuser

Password: .x8UP(ar.YZ2R)

Port: 22758 .

The product is available in the directory /becftp/OCEAN/SSS/SMOS/SouthernOcean/v1.0/L3/9day and in the ESA Open Science Catalog (https://opensciencedata.esa.int/products/sofresh-sea-surface-salinity/collection). The Argo profilers dataset from which the temporal correction is derived are available through the SEANOE webpage (htpps://www.seanoe.org/data/

00311/42182) or all the options to the access to data provided there.

*Author contributions.* All authors contributed significantly to this research. VGG, EO, AGE, CGH, and AT led the conceptualization and development of algorithms for product generation. Product validation was carried out by VGG, EO, AGE, MU, CG, and NH. AS, AN, ANG, and RC provided regional SOSE model data and contributed to the assessment of product's performance in comparison to the regional model. GA and YC supplied quality-controlled in situ data and contributed to discussions on the satellite product's performance in comparison to in

situ data. RS and DFP served as ESA technical officers. All authors actively participated in discussions regarding the product's quality and potential applications and reviewed the manuscript.

*Competing interests.* The contact author has declared that none of the authors has any competing interests.

*Acknowledgements.* This work has been carried out as part of the Southern Ocean Freshwater (SO-FRESH) project (AO/1-10461/20/I-NB), funded by the European Space Agency. It has been also supported by the H2020 CRices project (Climate Relevant interactions and feedbacks:

the key role of sea ice and Snow in the polar and global climate system)(Grant Agreement No. 101003826) and by the Spanish R&D projects INTERACT (PID2020-114623RB-C31) and EO4TIP (PID2023-149659OB-C21), which are funded by MCIN/AEI/10.13039/501100011033. We also acknowledge funding from the Spanish government through the 'Severo Ochoa Centre of Excellence' accreditation (CEX2019-000928-S). This work is a contribution to CSIC Thematic Interdisciplinary Platform PTI Teledetect. TSG data along the New Zealand/Antarctic transect are collected in the framework of several projects of the Programma Nazionale di Ricerche in Antartide- PNRA activities funded by

the Ministry of University and Research (MUR). The Marine Observatory of the Ross Sea is acknowledged. YC activities are performed in the framework of the ACCES project of the PNRA (PNRA 19 00032). The authors gratefully acknowledge the SGAI-CSIC for its assistance and help while using the DRAGO Supercomputer.

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
