# Peer review of "Satellite-based regional Sea Surface Salinity maps for enhanced understanding of freshwater fluxes in the Southern Ocean"

_Earth System Science Data, 2025_

## Author Comment (AC1)

**Review on paper ESSD-2025-212**

We would like to thank the Editor and the Reviewers for their useful insights and suggestions that have helped to improve the clarity of the manuscript. The detailed answers and the changes brought to our initial version are indicated below.

**Reviewer 1: Giuseppe M.R. Manzella**

**Citation: https://doi.org/10.5194/essd-2025-212-RC1**

**General comments:**

This is one of many papers that researchers clustered around BEC submit to journals and in particular to ESSD. This is a list (for sure not a complete one).

- (1) Improved BEC SMOS Arctic Sea Surface Salinity product v3.1 https://doi.org/10.5194/essd-14-307-2022
- (2) First SMOS Sea Surface Salinity dedicated products over the Baltic Sea https://doi.org/10.5194/essd-14-2343-2022
- (3) New SMOS SSS maps in the framework of the Earth Observation data For Science and Innovation in the Black Sea https://doi.org/10.5194/essd-2021-364
- (4) Nine years of SMOS Sea Surface Salinity global maps at the Barcelona Expert Center https://doi.org/10.5194/essd-13-857-2021

It is normal that 'general' algorithms need to be adapted to the particular marine environments, but the dispersion of applications in different papers is not useful to the researchers interested in using SMOS products for several small seas and analyzing the differences and possibly making further adjustments. The potential proliferation of articles for each single basin risks being of little use to research.

At the extreme north and south of the global ocean, important common problems are (1)Low sensitivity of brightness temperatures (TB) to salinity in cold waters, (2) Land–sea contamination (LSC) and ice–sea contamination (ISC), (3) Lack of sufficient in situ measurements. It would have been appropriate to discuss in a single publication the different peculiarities of the waters in these environments and what the appropriate solutions are. Same concept for the Baltic and Black Seas.

It is understood that in practice articles are written on the basis of collaborations that are built over time and therefore the practice of the authors is acceptable, but each article should highlight and point out the differences much more accurately. In particular, I would have expected a discussion after line 52 of the introduction on what has been done in the Arctic Sea and what needs to be done differently in the Southern Ocean.

As the reviewer notes, the previous BEC SSS products (1-3) and the product presented in the manuscript under review were all developed under different ESA-funded dedicated regional projects. In each of them, the BEC team collaborated with different leading experts to support the scientific exploitation of the respective SSS products. Besides that, the algorithms used to generate the SSS products have been specifically tailored to each region, as the methodological challenges of SSS retrieval and the geophysical characteristics vary significantly across different seas and oceans.

Even for the two polar products, there is a significant difference between the Arctic and Southern Oceans. In the Southern Ocean, the low variability of SSS implies the need to maximize the signal-to-noise ratio, particularly through enhanced Level 1 algorithms (i.e. at TB level).

Additionally, signal contamination near sea ice is a key aspect addressed for the first time in the Southern Ocean product by correcting SSS biases based on the distance from the sea ice edges.

This idea has been reinforced in the Introduction (Lines 43-47):

"The BEC team developed a specific product for the Arctic (Martínez et al., 2022). However, there is a significant difference between the Arctic and Southern Oceans. In the Southern Ocean, the SSS variability is notably lower than in the Arctic. This highlights the need of maximizing the signal-to-noise ratio, particularly through enhanced Level 1 algorithms. Additionally, signal contamination near sea ice is a key aspect to be specifically addressed in the Southern Ocean by correcting SSS biases based on the distance from the sea ice edges."

We thank the reviewer for pointing out that the key differences among the algorithms used to develop the various BEC SSS products should be clearly presented in the manuscript.

To address this, we have provided a detailed explanation of the main algorithmic differences between the products mentioned by the reviewer and the one presented in our study. A summary of all the differences is shown in the Table below. These details are included in Section 2.2.1, as we believe such technical content is more appropriate for a specialized section rather than the Introduction. Instead of inserting the table, which may be harder to interpret, we have chosen to describe the differences directly in the text to improve readability and comprehension.

| BEC product                                   | Input data      | Level 1 algorithms         |                                       | Level 2 algorithms
Debiased non-Bayesian |           |                                               |
|-----------------------------------------------|-----------------|----------------------------|---------------------------------------|---------------------------------------------|-----------|-----------------------------------------------|
|                                               |                 | ALL-LIC
EF Gkj
corr. | Nodal Sampling                        | Dielectric
constant
model             | Debiasing | SMOS-based
climatologies
(dependencies) |
| Global, v2.0
(Olmedo,
2021)             | ESA L1B
v620 | No                         | No                                    | Klein &
Swift                            | SSS       | (lon, lat, d, x, θ)                           |
| Arctic v3.1
(Martinez,
2022)            | ESA L1B
v620 | No                         | No                                    | Meissner &
Wentz                         | ТВ        | (lon, lat, d, xi, eta)                        |
| Baltic v1.0
(González-G
ambau,
2022) | ESA
Level 0  | Yes                        | No                                    | Modified
Meissner &
Wentz             | SSS       | (lon, lat, d, x, θ,T s )           |
| Black v1.0
(Olmedo,
2021)               | ESA
Level 0  | Yes                        | Yes
land-sea-sky mask              | Modified
Meissner &
Wentz             | SSS       | (lon, lat, d, x, θ,T s )           |
| Southern
Ocean v1.0                        | ESA
Level 0  | Yes                        | Yes
daily land-sea-ice-sky
mask | Modified
Meissner &
Wentz             | SSS       | (lon, lat, d, x, $\theta$ ,ice)               |

The following reference, which details the methodology for the generation of BEC Arctic SSS v3.1, has been added:

Martínez, J., Gabarró, C., Turiel, A., González-Gambau, V., Umbert, M., Hoareau, N., González-Haro, C., Olmedo, E., Arias, M., Catany, R., Bertino, L., Raj, R. P., Xie, J., Sabia, R., and Fernández, D.: Improved BEC SMOS Arctic Sea Surface Salinity product v3.1, Earth Syst. Sci. Data, 14, 307–323, https://doi.org/10.5194/essd-14-307-2022, 2022.

In the remaining part of the article, the work for the geophysical corrections and for the analyses of the various SST products is appreciable. The validation and intercomparison are also very appreciable.

The paper itself is well written and practically complete and can be published after minimal corrections.

**Specific comments:**

Introduction lines 14-15 - *understanding of processes influenced by upper-ocean salinity, including ice formation* – explain how it is possible to study the formation when SMOS resolution is large enough and the distance from the ice edge is more than 150 km, while within 150km there is high noise?

The manuscript states: "This product will significantly contribute to the understanding of processes influenced by upper-ocean salinity, including ice formation and melt, the reduction of Antarctic sea ice extent, and the opening of offshore polynyas".

When we mention "... ice formation and melt", we were referring to sea ice dynamics. We have seen in the study by Silvano et al., (2025) that the satellite SSS product presented in this manuscript provides essential evidence of the Southern Ocean's potential transition toward persistently-reduced sea ice coverage, even if that study uses the SSS located more than 200 km away from the sea ice edge.

We have modified this paragraph to be clearer in this aspect:

"This product will significantly contribute to the understanding of processes influenced by upper-ocean salinity, including sea ice dynamics, particularly, the reduction of Antarctic sea ice extent and the opening of offshore polynyas".

Figures 5, 7, 18 show very noisy areas near the continent. What does this mean for studies on the potential applications of the product: understanding of processes influenced by upper-ocean salinity, including ice formation and melt, the reduction of Antarctic sea ice extent, and the opening of offshore polynyas. It is not necessary to go into depth on the oceanographic issues, but to give a general indication of the limits of the product for the listed applications.

We would like to clarify that, in the case of Figure 5, each point of the map represents the typical SSS value for a specific acquisition condition (i.e., antenna position and orbit direction). These are "raw" SSS, before the application of any correction, and do not correspond to the final SSS maps. In the other two figures (7 and 18), which do correspond to final SSS maps, high SSS variability is shown in the first 100-150 km from the coast and ice margin, as pointed out by the reviewer.

We agree with the reviewer on the need of giving a general indication of the limits of the product for the listed applications. We have added the following text in the Conclusions section:

**Lines 407-411:**

"Therefore, part of the high variability observed within the first 100-150 km from the coast and ice margin reflects geophysical signals, and is also captured by the regional SOSE model. However, some of this variability could be attributed to residual contamination in the satellite-derived SSS product. Further research is required to understand which part of the variability is still residual contamination and then, to further enhance the quality near the boundaries, enabling more accurate description of critical processes occurring in close proximity to ice margins."

Despite current limitations in accuracy in the first 100-150 km to coast and ice margins, this satellite SSS product has proven capable of monitoring some of the recent changes in the Southern Ocean sea ice dynamics, as shown in the recent study by Silvano et al., (2025). The following text has been added to the manuscript.

**Lines 414-416:**

"This product has been used in the study by Silvano et al. (2025). Using these satellite observations, a marked increase of SSS across the circumpolar SO has been observed since 2015. This has weakened upper-ocean stratification, coinciding with the notably Antarctic sea ice retreat. Besides, increasing salinity also contributed to the reappearance of the Maud Rise polynya in the Weddell Sea, during the winters of 2016-2017. Future work will focus on further evaluating the quality of SSS in the polynyas."

The following reference has also been added to the manuscript:

Silvano, A., Narayanan, A., Catany, R., Olmedo, E., González-Gambau, V., Turiel, A., Sabia, R., Mazloff, M.R., Spira, T., Haumann, F.A., Naveira Garabato, A.C.: Rising surface salinity and declining sea ice: A new Southern Ocean state revealed by satellites, Proc. Natl. Acad. Sci. U.S.A. 122 (27) e2500440122, https://doi.org/10.1073/pnas.2500440122, 2025.

**Reviewer 2**

**Citation: https://doi.org/10.5194/essd-2025-212-RC2**

**General comments:**

This manuscript describes a new sea surface salinity (SSS) product for the Southern Ocean derived from SMOS observations, incorporating algorithmic improvements aimed at enhancing retrieval accuracy near sea ice margins. The dataset spans over a decade, and the validation suggests reasonable agreement with independent in situ observations in open waters. The topic is relevant to the ESSD readership, and the dataset could be a useful resource for studies related to polar oceanography and sea-ice interactions.

There are several important aspects that require further clarification and improvement. First, the manuscript does not sufficiently distinguish the presented dataset from previously published SSS products, including those already available in ESSD. A more detailed comparison is needed to highlight what has been improved in terms of retrieval methodology, spatial and temporal resolution, coverage, or accuracy. It would be helpful to clarify which known limitations in prior products this dataset addresses.

**We refer the reviewer to our responses to Reviewer 1, where a similar comment regarding the differences between the presented dataset and previously published BEC SSS products has already been addressed.**

The description of the data processing chain, particularly the modified retrieval algorithms, should be expanded. Although some methods have been previously published, ESSD readers should be able to understand the core processing logic and improvements without needing to refer to external literature. Key steps—especially those tailored to address the challenges of SSS retrieval in cold, ice-influenced waters—should be described in sufficient detail to ensure transparency and reproducibility.

We agree that it is important for readers to understand the core data processing logic. However, due to the complexity of the processing chain, this is not fully achievable without referring to

external literature that explains the underlying methodologies. To enhance clarity, we have aimed to find a balance between referencing previous works and providing a self-contained explanation within the manuscript. In particular, we have expanded the explanations about how the Nodal Sampling and the Debiased Non-Bayesian retrieval have been modified to address the specific challenges of SSS retrieval in cold, ice-influenced waters. The main changes are highlighted in the Section 2.2 Algorithm developments. Further details are provided to the specific comments below.

The link provided for accessing the dataset is currently not functioning. The authors should ensure that the data are fully accessible via a persistent and reliable repository, in compliance with ESSD's data availability requirements. The dataset should be accompanied by appropriate documentation and metadata.

The reviewer is right, and we apologize for the inconvenience. We experienced issues with the BEC FTP server during the period in which the manuscript was under review. We have changed the dataset to a persistent and reliable repository. The product with their metadata and associated documentation can be directly downloaded from the directory /becftp/OCEAN/SSS/SMOS/SouthernOcean/v1.0/L3/9day, using the following credentials:

Host: sftp://eodata-bec.icm.csic.es Username: ftpuser Password: .x8UP(ar.YZ2R) Port: 22758

This information has been updated in the manuscript, in the section Data availability.

In addition, the manuscript references other datasets used in the validation and development process, but no access links or citation details are provided. All datasets mentioned or used should be properly cited and made accessible to ensure full transparency and reproducibility.

Accordingly, we have carefully reviewed the citation details and access links to all the datasets used in the product development and in the intercomparison and validation. These are the modifications we have done in the manuscript:

Auxiliary data for SSS retrieval (section 2.1):

- Sea Surface Temperature. A reference to the GHRSST Level 4 MUR Global Foundation SST Analysis (v4.1) product has been added (NASA/JPL, 2015).
- Annual salinity climatology. The previous link is no longer active. A reference to the dataset has been added (Levitus et al., 2014).

Data for SSS filtering and correction (section 2.1):

• Argo floats. The following sentence has been added: These measurements can be downloaded from ftp://ftp.ifremer.fr/ifremer/argo.

Datasets for intercomparison and validation (section 3.1):

- Thermosalinograph (TSG) data by the Università degli Studi di Napoli Parthenope. The reference for the Pacific dataset has been added.
- TSG data by the Astrolabe vessel. The reference has been changed by a more updated one (Morrow and Kestenare, 2014). The following text has been added: It is provided by the Survostral project and available at https://www.legos.omp.eu/survostral/data-products/tsg-sss-sst/.

- Barcelona World Races (BWR) 2011, 2015 and Vendée Globe 2020. A recently published paper on the analysis of these in situ datasets has been added (Hernani et al., 2025). Only the 2015 dataset is publicly available.
- The GLORYS12V1 product. The following text has been added: This product is available https://data.marine.copernicus.eu/product/GLOBAL\_MULTIYEAR\_PHY\_001\_030/files? subdataset=cmems\_mod\_glo\_phy\_my\_0.083deg\_P1M-m\_202311.
- The B-SOSE model iteration 135. The following text has been added: We use the Iteration 135 (available at https://sose.ucsd.edu/SO6/ITER135/)

The following references have been added:

- Levitus, S., Boyer, T. P., García, H. E., Locarnini, R. A., Zweng, M. M., Mishonov, A. V., Reagan, J. R., Antonov, J. I., Baranova, O. K., Biddle, M., Hamilton, M., Johnson, D. R., Paver, C. R., and Seidov, D.: World Ocean Atlas 2013 (NCEI Accession 0114815). https://doi.org/https://doi.org/10.7289/v5f769gt, https://www.ncei.noaa.gov/data/oceans/woa/WOA13/DATAv2/salinity/netcdf/decav/0.25 /woa13\_decav\_s00\_04v2.nc, 2014.
- Morrow, R. and Kestenare, E. Nineteen-year changes in surface salinity in the Southern Ocean south of Australia. J. Mar. Sys., 129:472–483, January 2014. doi: 10.1016/j.jmarsys. 2013.09.011.
- NASA/JPL: GHRSST Level 4 MUR Global Foundation Sea Surface Temperature Analysis (v4.1), https://doi.org/10.5067/GHGMR-4FJ04, http://podaac.jpl.nasa.gov/dataset/MUR-JPL-L4-GLOB-v4.1, 2015.
- Hernani, M., Werner-Pelletier, N., Umbert, M., Hoareau, N., Olivé-Abelló, A., and Salat, J.: Inter-annual hydrographic changes in the Southern Ocean: analysis of Vendée Globe Race and Barcelona World Race data, Antarctic Science, p. 1–18, https://doi.org/10.1017/S0954102025100138, 2025.

**Specific comments:**

Multiple algorithm steps labeled as "Main algorithm changes" are mentioned in Figure

 but the manuscript does not provide detailed explanations of these changes
 individually in the main text.

Three steps are labeled as "Main algorithm changes" in Figure 1. The explanations of "Computation of dynamic sea-ice-land mask" and "Application of NSv3 (dynamic mask)" were detailed in the Section 2.2.2 Reduction of TB radiometric errors. The explanation of "Computation of SMOS climatologies, depending on acquisition conditions and sea ice distance" was detailed in the section 2.2.3 Reduction of SSS systematic errors.

To clarify this point, we have updated the titles of sections 2.2.2 and 2.2.3 to align with Figure 1. Additionally, we have expanded the explanations of the main algorithm changes for better clarity. Further details are provided in Specific comments 3 and 4.

All figures in the manuscript lack sufficient clarity, making it difficult to interpret the visualized information. Additionally, some figures (e.g., Figures 5–7, 10-15, and 18-19) and Tables 1 and 2 do not include units, particularly in color bars and value columns. Please revise to improve figure resolution and ensure all numerical data are properly labeled with units.

Practical Salinity is defined on the Practical Salinity Scale of 1978. Note that Practical Salinity is a unit-less quantity. We have followed the recommendation in (Section 2.3): https://www.teos-10.org/pubs/TEOS-10\_Manual.pdf

We have added in the manuscript the following sentence in Lines 190-191, since it is the first time SSS values are presented:

"Note that the SSS values presented throughout the paper refer to Practical Salinity, expressed on the Practical Salinity Scale of 1978 (PSS-78), which is a unitless quantity."

Most figures in the manuscript have been improved.

3. In line 144, the term "land-sea-sky mask" is mentioned, but its definition and implementation are unclear. Later, Figure 7 introduces a "land-sea-ice mask", yet the relationship between these two masks is not explained. Please provide a clearer and more detailed description of how the land-sea-sky mask is constructed and clarify whether it differs from the land-sea-ice mask.

The reviewer is right. The definition and implementation of the mask were unclear in the previous version of the manuscript.

For the Southern Ocean, we use a land–sea–ice–sky mask when applying the nodal sampling. In developing the Black Sea SSS product, a land–sea–sky mask was introduced for the first time to effectively prevent artificial contamination near coastlines and the Earth–sky horizon due to the mixing of different pixel types (land/ocean/sky) during the computation of the Laplacian on the original grid. A similar issue was observed in the Southern Ocean at ocean–ice boundaries. When sea ice concentration (SIC) varies significantly, the associated brightness temperatures also change significantly, resulting in artificially increased ocean TB values near these transitions. To address this, we incorporate for the first time a daily SIC product to create a dynamic, daily land–sea–ice–sky mask.

We have improved the description on how the mask is built and applied during the refinement of the nodal points selection in the coarse grid (see changes implemented in Section 2.2.2).

4. The manuscript briefly mentions the "DNB-ice" approach as the preferred method for generating the BEC SO SSS product, but it lacks a clear description of how it differs from the standard DNB and DNB-SST methods. Please consider expanding this explanation.

The only difference among the three tested versions of the DNB lies in how the raw SSS are classified to compute the SMOS-based climatologies for fixed acquisition conditions,  $\gamma$ , which is then used to correct the systematic spatial biases on SSS:

- In the standard DNB: The classification depends on latitude, longitude, orbit direction, across-track distance to the center of the swath and incidence angle. The acquisition conditions are defined as a 5-tuple *γ*=(φ,λ,d,x,*θ*).
- In the DNB-SST: The classification depends on the same conditions as the standard DNB but a new variable is introduced: the SST. The acquisition conditions are defined as a 6-tuple γ=(φ,λ,d,x,θ,Ts).

 DNB-ice: The classification depends on the same conditions as the standard DNB but a new variable is introduced instead of the SST: the distance from sea ice edge. The acquisition conditions are defined as a 6-tuple, *γ*=(φ,λ,d,x,θ,lce).

We have improved the description on how the DNB-ice differs from the standard DNB (Olmedo et al., 2017, 2021) and from the DNB-SST (González-Gambau et al., 2022). Changes are highlighted in Section 2.2.3.

5. While the section 3.3.2 compares satellite SSS with both marine mammal and TSG data, it does not discuss potential sampling differences or biases between the two in situ data sources, which could affect the interpretation of near-coast and near-ice performance. Please consider addressing this aspect.

Thank you for the suggestion. We have added a discussion in section 3.3.2 addressing potential sampling differences between the two in situ datasets.

The following text has been added in the manuscript:

**Lines 353-360**

"It is important to note that the two in situ datasets exhibit several sampling differences. Their spatial coverage differs significantly. TSG data are mostly regularly distributed along the meridional transects followed by the research vessels across various sectors of the SO. Although concentrated on very specific longitudes, they provide a relatively broad circumpolar representation of in situ conditions and occasionally extend into more southerly latitudes (e.g., beyond 65°S in the Ross Sea) exceeding the spatial extent of the marine mammals dataset. In contrast, the marine mammals dataset is largely confined to a single sector of the SO, with sampling mainly concentrated near the sea ice edge. Moreover, there is a marked difference in the temporal distribution of the two datasets. Observations from marine mammals are predominantly acquired during the winter months, whereas TSG data are primarily collected during the summer."

**Lines 368-370**

"Differences in statistics may stem from the previously mentioned sampling differences. In both cases, the largest errors occur within the first 100-150 km from the sea ice edge and/or the coastline, as shown in Fig. 16."